# Truthful High Dimensional Sparse Linear Regression

**Liyang Zhu**[1], **Amina Manseur**[1] , **Meng Ding**[2], **Jinyan Liu**[3] , **Jinhui Xu**[2] , **Di Wang**[1]

[1]PRADA Lab, King Abdullah University of Science and Technology
[2]State University of New York at Buffalo
[3]Beijing Institute of Technology
{liyang.zhu, amina.manseur, di.wang}@kaust.edu.sa
{mengding, jinhu}@buffalo.edu
jyliu@bit.edu.cn

## Abstract

We study the problem of fitting the high dimensional sparse linear regression model with sub-Gaussian covariates and responses, where the data are provided by strategic or self-interested agents (individuals) who prioritize their privacy of data disclosure. In contrast to the classical setting, our focus is on designing mechanisms that can effectively incentivize most agents to truthfully report their data while preserving the privacy of individual reports. Simultaneously, we seek an estimator which should be close to the underlying parameter. We attempt to solve the problem by deriving a novel private estimator that has a closed-form expression. Based on the estimator, we propose a mechanism which has the following properties via some appropriate design of the computation and payment scheme: (1) the mechanism is $(o(1), O(n^{-\Omega(1)}))$-jointly differentially private, where $n$ is the number of agents; (2) it is an $o(\frac{1}{n})$-approximate Bayes Nash equilibrium for a $(1 - o(1))$-fraction of agents to truthfully report their data; (3) the output could achieve an error of $o(1)$ to the underlying parameter; (4) it is individually rational for a $(1 - o(1))$ fraction of agents in the mechanism; (5) the payment budget required from the analyst to run the mechanism is $o(1)$. To the best of our knowledge, this is the first study on designing truthful (and privacy-preserving) mechanisms for high dimensional sparse linear regression.

## 1 Introduction

One fundamental learning task is estimating the linear regression model, with a wide array of applications ranging from statistics to experimental sciences like medicine [7, 23] and sociology [36]. These studies typically assume that analysts have high-quality data, which is essential to the success of the model. However, real-world data, such as medical records and census surveys, often contain sensitive information sourced from strategic, privacy-concerned individuals. In this case, data providers (agents)[1] may be disinclined to reveal their data truthfully, potentially jeopardizing the accuracy of model estimation. Therefore, in contrast to conventional statistical settings, it becomes imperative to model the utility functions of individuals and engineer mechanisms that can concurrently yield precise estimators, safeguard the privacy of individual reports, and encourage the majority of individuals to candidly disclose their data to the analyst.

The problem involves two intertwined components: data acquisition and privacy-preserving data analysis. The analyst must strategically compensate agents for potential privacy violations, considering the alignment of their reported data with both the statistical model and their peers' contributions, while minimizing the payment budget. Additionally, the analyst must perform privacy-preserving computations to accurately learn the underlying model. As agents cannot refine their reports after seeing their payment, the interaction is designed to be completed in one round, creating a trade-off between estimator accuracy and the total payment budget needed for participant cooperation.

---

[1]In this paper, individuals, data providers, and participants are the same and all represent agents.

In recent years, there has been a line of work studying truthful and privacy-preserving linear models such as [2, 37, 14, 38]. However, due to the complex nature of the problem, all of them only consider the low dimension case, where the dimension of the feature vector (covariate) is much lower than the sample size, i.e., they need to assume the dimension is a constant. In practice, we always encounter the high dimensional sparse case where the dimension of the feature vector is far greater than the sample size but the underlying parameter has an additional sparse structure. While the (high dimensional sparse) linear model has been widely studied in statistics and privacy-preserving machine learning [39, 24, 3, 52, 61, 53, 49, 28], to the best of our knowledge, there is no previous study on truthfully and privately estimating high dimensional sparse linear models due to some intrinsic challenges of the problem (see Section C in Appendix for details). Therefore, a natural question to ask is:

*Can we fit the high dimensional sparse linear regression and design mechanism which incentivizes most agents and truthfully report their data and preserves the privacy of the individuals?*

In this paper, we answer the question in the affirmative via providing the first study on the trade-off between privacy, the accuracy of the estimator, and the total payment budget for high dimensional sparse linear models where the dimension $d$ can be far greater than the sample size $n$, while the sparsity $k$ and $\log d$ are far less than $n$ can be considered as constants. Specifically, for the privacy-preserving data analysis part, we adopt the definition of Joint Differential Privacy (JDP) [31] to protect individuals' data and develop a novel closed-form and JDP estimator for sparse linear regression, which is significantly different from the low dimension case and can be applied to other problems. Moreover, we develop a general DP estimator for the $\ell_p$-sparse covariance matrix with $p \in [0, 1]$ as a by-product, which extends the previous results on the $\ell_0$-sparse case. Based on our JDP estimator, via the peer prediction method, we then provide a payment mechanism that can incentivize almost all participants to truthfully report their data with a small amount of the total payment budget.

In detail, our mechanism has the following properties. (1) The mechanism preserves privacy for individuals' reported data, i.e., the output of the mechanism is $(o(1), O(n^{-\Omega(1)}))$-JDP, where $n$ is the number of agents. (2) The private estimator of the mechanism is $o(1)$-accurate, i.e., when the number of agents increases, our private estimator will be sufficiently close to the underlying parameter. (3) The mechanism is asymptotically truthful, i.e., it is an $o(\frac{1}{n})$-approximate Bayes Nash equilibrium for a $(1 - o(1))$-fraction of agents to truthfully report their data. (4) The mechanism is asymptotically individually rational, i.e., the utilities of a $(1 - o(1))$-fraction of agents are non-negative. (5) The mechanism only requires $o(1)$ payment budget, i.e., when the number of participants increases, the total payment tends to zero.

## 2 Preliminaries

**Notations.** Given a matrix $X \in \mathbb{R}^{n \times d}$, let $x_i^T$ be its $i$-th row and $x_{ij}$ (or $[X]_{ij}$) be its $(i, j)$-th entry (which is also the $j$-th element of the vector $x_i$). For any $p \in [1, \infty]$, $\|X\|_p$ is the $p$-norm, i.e., $\|X\|_p := \sup_{y \neq 0} \frac{\|Xy\|_p}{\|y\|_p}$, and $\|X\|_{\infty,\infty} = \max_{i,j} |x_{ij}|$ is the max norm of matrix $X$. For an event $A$, we let $I[A]$ denote the indicator, i.e., $I[A] = 1$ if $A$ occurs, and $I[A] = 0$ otherwise. The sign function of a real number $x$ is a piece-wise function which is defined as $\mathrm{sgn}(x) = -1$ if $x < 0$; $\mathrm{sgn}(x) = 1$ if $x > 0$; and $\mathrm{sgn}(x) = 0$ if $x = 0$. We also use $\lambda_{\min}(X)$ to denote the minimal eigenvalue of $X$. For a sub-Gaussian random variable $X$, its sub-Gaussian norm $\|X\|_{\psi_2}$ is defined as $\|X\|_{\psi_2} = \inf\{c > 0 : \mathbb{E}[\exp(\frac{X^2}{c^2})] \leq 2\}$ (see Appendix E for more preliminaries).

### 2.1 Problem Setting

We consider the problem of sparse linear regression in the high dimensional setting where $d \gg n$. Suppose that we have a data universe $\mathcal{D} = \mathcal{X} \times \mathcal{Y} \subseteq \mathbb{R}^d \times \mathbb{R}$ and $n$ agents in the population, where each agent $i$ has a feature vector $x_i \in \mathcal{X}$ and a response variable $y_i \in \mathcal{Y}$ (we denote $D_i = (x_i, y_i)$ and $D$ as the whole dataset). We assume that $\{(x_i, y_i)\}_{i=1}^n$ are i.i.d. sampled from a sparse linear regression model, i.e., each $(x_i, y_i)$ is a realization of the sparse linear regression model $y = \langle \theta^*, x \rangle + \zeta$, where the distribution of $x$ has mean zero, $\zeta$ is some randomized noise that satisfies $\mathbb{E}[\zeta|x] = 0$, and $\theta^* \in \mathbb{R}^d$ is the underlying model estimator with sparsity assumption $\|\theta^*\|_0 \leq k$. In the following, we provide some assumptions related to the model. See Section A in Appendix for a table of notations.

**Assumption 1.** *We assume $\|\theta^*\|_2 \leq 1$. Moreover, for the covariance matrix of $x$, $\Sigma$, there exist $\kappa_\infty$ and $\kappa_x$ such that $\|\Sigma w\|_\infty \geq \kappa_\infty \|w\|_\infty, \forall w \neq 0$ and $\|\Sigma^{-\frac{1}{2}} x\|_{\psi_2} \leq \kappa_x$.* [2]

---

[2]To make our results comparable to the previous results, we assume $\kappa_\infty$ and $\kappa_x$ are constants for simplicity.

These assumptions have been commonly adopted in some relevant studies such as [52, 14, 3]. Next, we present the assumptions made on the covariate vectors $x_i$ and response variable $y_i$.

**Assumption 2.** *We assume that the covariates (feature vectors) $x_1, x_2, \cdots, x_n \in \mathbb{R}^d$ are i.i.d. (zero-mean) sub-Gaussian random vectors with variance $\frac{\sigma^2}{d}$ with $\sigma = O(1)$. [3] The responses $y_i$ are i.i.d. (zero-mean) sub-Gaussian random variables with variance $\sigma^2$ with $\sigma = O(1)$.*

In our setting, the agents are self-interested or strategic. They concerned about their privacy and can misreport their responses $\{y_i\}_{i=1}^n$ but not their features $\{x_i\}_{i=1}^n$. This means the features are directly observable but the response (e.g., during physical examination) is unverifiable. We denote $\hat{D} = \{\hat{D}_i = (X_i, \hat{y}_i)\}_{i=1}^n$ the reported dataset where $\hat{y}_i = \sigma_i(D_i)$ is the reporting strategy adopted by agent $i$. Specifically, each agent is characterized by a privacy cost coefficient $c_i \in \mathbb{R}_+$. Higher $c_i$ indicates the agent $i$ is concerned more about the privacy violation.

Apart from the agents, there is an analyst who seeks an accurate estimator $\bar{\theta}$ of $\theta^*$ based on the reported data and needs to construct a payment rule $\pi : \mathcal{D}^n \to \Pi^n$ that encourages the truthful participation, i.e., reveal their data truthfully to the agent. Misreporting the response $y_i$ will result in a decrease in the payment received $\pi_i$. The analyst will thus design a mechanism $\mathcal{M}$ takes the reported dataset $\hat{D} = \{\hat{D}_i = (X_i, \hat{y}_i)\}_{i=1}^n$ as input and outputs an estimator $\bar{\theta}$ of $\theta^*$ and a set of non-negative payments $\{\pi_i\}_{i=1}^n$ for each agent. This mechanism will in turn satisfy some privacy guarantee for the reports provided by agents. Moreover, the desired mechanism must constrain the payments to an asymptotically small budget. All of the above discussion depends upon the rationality of each agent.

## 2.2 Differential Privacy

Due to space limit, we postpone all the relevant definitions of DP to the appendix, readers can refer to Section. A.1. In our case, DP requires that all outputs, including the payments allocated to agents, are insensitive to each agent's input. This requirement is quite strict, as the payment to each agent is not shared publicly or with other agents. Therefore, instead of using the original DP, we consider a relaxation known as *joint differential privacy* (JDP) [31].

## 2.3 Utility of Agents

Based on the definition of JDP, we introduce in this section our model of agent utilities. We assume that each agent $i$ is characterized by her privacy cost parameter $c_i \in \mathbb{R}_+$ representing how she is concerned about the privacy violation in the case she truthfully reports $y_i$ to the analyst. We also introduce her privacy cost function $f_i(c_i, \varepsilon, \delta)$ which measures the cost she incurs when her response $y_i$ is used in an $(\varepsilon, \delta)$-Joint Differential Private Mechanism. Considering the payment $\pi_i$ of agent $i$ and her privacy cost function $f_i(c_i, \varepsilon, \delta)$, we denote by $u_i = \pi_i - f_i(c_i, \varepsilon, \delta)$, her utility function that represents the utility she gets when she reports her response $y_i$ truthfully to the analyst. Following the previous works, we assume that all functions $f_i$ are bounded by a function of $c_i$ and $\varepsilon$, increasing with $\varepsilon$.

**Assumption 3.** *The privacy cost function of each agent $i$ satisfies $f_i(c_i, \varepsilon, \delta) \leq c_i(1 + \delta)\varepsilon^3$*

Larger values of $\varepsilon$ and $\delta$ imply weaker privacy guarantees, which means the privacy cost of an agent becomes larger. Thus, it is natural to let $f_i$ be bounded by a component-wise increasing function, which can be denoted by $F(c_i, \varepsilon, \delta)$. Here $F(c_i, \varepsilon, \delta) = c_i(1 + \delta)\varepsilon^3$. In [14], the authors consider the case where the response is bounded and $\delta = 0$, they assume $f_i(c_i, \varepsilon, \delta) \leq c_i\varepsilon^2$, i.e., $F(c_i, \varepsilon, \delta) = c_i\varepsilon^2$. However, since our Assumption 2 on the distribution of the response is more relaxed, we need stronger assumptions regarding the privacy cost function and the distribution of privacy cost coefficients.

We assume that the privacy cost parameters are also random variables, sampled from a distribution $\mathcal{C}$. It is intuitive to believe that the privacy cost $c_i$ for each individual does not reflect the privacy costs incurred by other individuals. Also, we allow $c_i$ to be correlated with its corresponding data sample $D_i$. Therefore, we make the following assumption.

**Assumption 4.** *Given $D_i$, $(D_{-i}, c_{-i})$ is conditionally independent of $c_i$, for each $i \in [n]$ :*

$$p(D_{-i}, c_{-i}|D_i, c_i) = p(D_{-i}, c_{-i}|D_i, c_i')$$

---

[3] In order to make our result comparable to previous work on linear regression with bounded covariates, we assume the variance proxy is $\sigma^2/d$ so that with high probability $\|x_i\|_2$ is bounded by some constant.

*where $c_{-i}$ represents the set of privacy costs excluding the privacy cost of agent $i$.*

In addition, we make the same assumption on the tail of $\mathcal{C}$ as in [37] that the probability distribution of $c_i$ has exponential decay. This assumption is essential for providing a bound on the threshold value $\tau_{\alpha,\beta}$ of truthful reporting, which will be explained in the following section.

**Assumption 5.** *There exists some constant $\lambda > 0$ such that the conditional distribution of the privacy cost coefficient satisfies $\inf_{D_i} \mathbb{P}_{c_i \sim p(c_i|D_i)}(c_i \leq \tau) \geq 1 - e^{-\lambda \tau}$.*

### 2.4 Truthful Mechanism Properties

Following the previous work, we propose to design mechanisms that satisfy the following properties : (1) truthful reporting is an equilibrium; (2) the private estimator output should be accurate; (3) the utilities of almost all agents are ensured to be non-negative; and (4) the total payment budget required from the analyst to run the mechanism is small. These concepts will be measured and evaluated using the framework of a multiagent, one-shot, simultaneous-move symmetric Bayesian game, where the behavior of the agents is modeled. Due to space limit, we provide rigorous definitions of the truthful mechanism properties to the Appendix. Readers can refer to Section A.2 for details.

## 3 Main Results

In this section, we will design truthful and private mechanisms for our problem. Generally speaking, our method consists of two components: a closed-form JDP estimator for high dimensional sparse linear regression and a payment mechanism. In the following, we first introduce our private estimator for the original dataset $D$.

### 3.1 Novel Efficient Private Estimator

As mentioned in Section 1, the one-round communication constraint necessitates a closed-form estimator. For this reason, in the low dimension setting, previous methods did not favour LASSO as there is no closed-form solution and thus also cannot be used. [14, 37] are motivated by the closed-form solution of the ordinary least square (OLS) or ridge regression (See Section C for detailed discussion). However, it is well-known that for high dimensional sparse setting these two estimators come with less favorable guarantees when compared to the LASSO. When employed as a regularization term, the $\ell_2$-norm does not encourage sparsity to the same extent as the $\ell_1$-norm. Thus, all previous methods for truthful linear regression cannot be applied to our problem.

In the following, we derive a new estimator based on a variant of the classical OLS estimator, tailored for the high dimensional sparse setting. Our aim is twofold: first, to achieve convergence rates similar to the LASSO, and second, to exploit the inherent sparsity of the model.

We initiate our analysis with an initial, albeit unrealistic "scaffolding" assumption that the inverse of the empirical covariance matrix $(\hat{\Sigma}_{XX})^{-1}$ exists with $\hat{\Sigma}_{XX} = \frac{1}{n} \sum_{i=1}^{n} x_i x_i^T$, which will be removed in the course of our analysis. Intuitively, our goal is to find a sparse estimator that is close to OLS, i.e., $\arg\min_\theta \|\theta - \hat{\Sigma}_{XX}^{-1} \hat{\Sigma}_{XY}\|_2^2$, s.t. $\|\theta\|_0 \leq k$, whose $\ell_1$ convex relaxation of the $\ell_0$ constraint is

$$\arg\min_\theta \|\theta - \hat{\Sigma}_{XX}^{-1} \hat{\Sigma}_{XY}\|_2^2 + \lambda_n \|\theta\|_1 \tag{1}$$

with some $\lambda_n > 0$.

Although (1) is very similar to LASSO, fortunately, the above minimizer is just the proximal operator on OLS: $\text{Prox}_{\lambda_n\|\cdot\|_1}(\hat{\Sigma}_{XX}^{-1} \hat{\Sigma}_{XY})$, which has a closed form expression. Since the proximal operator is separable with respect to both vectors, $\theta$ and $\hat{\Sigma}_{XX}^{-1} \hat{\Sigma}_{XY}$, we have $(\text{Prox}_{\lambda_n\|\cdot\|_1}(\hat{\Sigma}_{XX}^{-1} \hat{\Sigma}_{XY}))_i = \text{sgn}((\hat{\Sigma}_{XX}^{-1} \hat{\Sigma}_{XY})_i) \max\{|(\hat{\Sigma}_{XX}^{-1} \hat{\Sigma}_{XY})_i| - \lambda_n, 0\}$. And the optimal solution $\hat{\theta}$ satisfies:

$$\hat{\theta} = S_{\lambda_n}(\hat{\Sigma}_{XX}^{-1} \hat{\Sigma}_{XY}), \tag{2}$$

where for a given thresholding parameter $\lambda$, the element-wise *soft-thresholding* operator $S_\lambda : \mathbb{R}^d \mapsto \mathbb{R}^d$ for any $u \in \mathbb{R}^d$ is defined as the following: the $i$-th element of $S_\lambda(u)$ is defined as $[S_\lambda(u)]_i = \text{sgn}(u_i) \max(|u_i| - \lambda, 0)$.

A key insight is that this operation is equivalent to embedding the classical OLS estimator within the sparsity constraint. This secures $\ell_2$-norm consistency which does not hold for neither the ridge

regression nor the original OLS estimator. The $\ell_1$ regularization serves to minimize the structural complexity of the parameter under constraints. Importantly, the estimator above is available in closed-form.

Our next focus centers on the privatization scheme. Prior work on truthful linear regression [14, 37] employed the output perturbation method, which adds some noise to the closed-form estimator. The noise should be calibrated by the $\ell_2$-sensitivities of their non-private estimators to ensure DP, which depend on $\text{Poly}(\sqrt{d/n})$. This is unacceptable when $d \gg n$. Should we adopt the same strategy to analyze the sensitivity of $S_{\lambda_n}(\hat{\Sigma}_{XX}^{-1}\hat{\Sigma}_{XY})$, we inevitably encounter the estimation error of $\hat{\Sigma}_{XX}^{-1}$. This error will cause the noise to have an unavoidable dependency on $\text{Poly}(\sqrt{d/n})$ [46]. Given the previous discussion, we propose to inject Gaussian noise separately to $\hat{\Sigma}_{XX}$ and $\hat{\Sigma}_{XY}$. It is notable that while similar methods have been used in DP statistical estimation [40, 48], it has not been used in truthful linear regression. This is due to that in our problem, the data is misreported, which makes the utility analysis more difficult. We will discuss it in Section 4.1.

For the term $\hat{\Sigma}_{XX}$, we apply $\ell_2$-norm clipping so that the sensitivity of clipped version $\hat{\Sigma}_{\bar{X}\bar{X}}$ is irrelevant to $d$. For the term $\hat{\Sigma}_{XY}$, since the covariate and the response $y_i$ are sub-Gaussian. Therefore, clipping operation becomes necessary. We shrink each coordinate of $x_i$ via parameters $\tau_x$, i.e., for $j \in [d]$ and $\tilde{x}_i = \text{sgn}(x_i)\min\{|x_i|, \tau_x\}$. We also perform the element-wise clipping on the response $y_i$ by $\tau_y$. Then the server aggregates these terms $\bar{x}_i\bar{x}_i^T$ and $\tilde{x}_i\tilde{y}_i$ and separately add Gaussian noises to $\hat{\Sigma}_{\bar{X}\bar{X}}$ and $\hat{\Sigma}_{\tilde{X}\tilde{Y}}$. The noisy version is denoted by $\dot{\Sigma}_{\bar{X}\bar{X}}$ and $\dot{\Sigma}_{\tilde{X}\tilde{Y}}$.

We now give a second thought to the estimation of the covariance matrix. As mentioned earlier, we used the scaffolding assumption on the invertibility of $\hat{\Sigma}_{XX}$ matrix, which does not hold as the matrix is rank-deficient if $d \gg n$. To mitigate this issue, we need to impose additional assumptions and here switch to the sparsity assumption introduced below on the structure of the covariance matrix, which has been widely studied previously such as [4, 5].

**Assumption 6.** *We assume that $\Sigma \in \mathcal{G}_q(s)$ for some $0 \leq q < 1$, where $\mathcal{G}_q(s) = \{\Sigma = (\sigma_{xx^T,ij})_{1\leq i,j\leq d} : \max_i \sum_{j=1}^d |\sigma_{xx^T,ij}|^q \leq s, \forall j \in [d]\}$ is the parameter space of $s$-approximately sparse covariance matrices.*

It is notable that the sparse covariance assumption is commonly adopted in the previous work on high dimensional estimation [60, 9]. To the best of our knowledge, the only work that considers the private sparse covariance matrix estimation is [51]. Unfortunately, [51] only considers the case where $q = 0$, which is a special case of Assumption 6 and we cannot trivially apply their method to our setting. Also, as we will discuss in Remark 3, even if the assumption does not hold, all of our theoretical results still hold if the sample size is large enough.

Directly using the perturbed covariance matrix will be insufficient to exploit the sparsity structure. In fact, it can be readily seen that $\|\dot{\Sigma}_{\bar{X}\bar{X}} - \Sigma\|_2 \leq O(\frac{\sqrt{d}}{n\epsilon})$, which is quite large under the high dimensional setting. To see why $\dot{\Sigma}_{\bar{X}\bar{X}}$ will introduce a large error under Assumption 6, we observe that some of its entries are quite large which makes $|\dot{\sigma}_{\bar{x}\bar{x}^T,ij} - \sigma_{xx^T,ij}|$ large for some $i, j$. Thus, we need to develop a new private estimator for (approximately) sparse covariance matrices as a valuable by-product. By Lemma 24 and 11 in the Appendix, we can get the following, with high probability, for all $1 \leq i, j \leq d$, $|\dot{\sigma}_{\bar{x}\bar{x}^T,ij} - \sigma_{xx^T,ij}| \leq \tilde{O}(\frac{\sqrt{\ln\frac{1}{\delta}}}{n\varepsilon})$. However, under the sparsity assumption, there will be two cases: (1) If $\sigma_{xx^T,ij}$ is small enough, then maybe the zero entry could have a smaller error than $\dot{\sigma}_{\bar{x}\bar{x}^T,ij}$ since the noise is quite large. (2) If $\sigma_{xx^T,ij}$ is large, then the original $\dot{\sigma}_{\bar{x}\bar{x}^T,ij}$ could be better. Motivated by the above observation, we perform a *hard-thresholding* operation on each $\dot{\sigma}_{\bar{x}\bar{x}^T,ij}$. This method takes advantage of this sparsity assumption by first estimating the sample covariance matrix, and then setting all entries with absolute values below a certain threshold *Thres* to 0. To be more specific, it filters out $\dot{\sigma}_{\bar{x}\bar{x}^T,ij}$ smaller than *Thres* and sets them to 0 while keeping those larger than *Thres* unchanged. This effectively shrinks the magnitude of the perturbed covariance matrix and thus lowering the error since some small $\sigma_{xx^T,ij}$ correspond to large $\dot{\sigma}_{\bar{x}\bar{x}^T,ij}$. After the hard-thresholding operation, we denote the resulting matrix by $\ddot{\Sigma}_{\bar{X}\bar{X}}$ and it is invertible with high probability as shown below.

---
**Algorithm 1** $(\varepsilon, \delta)$-DP Algorithm for Sparse Linear Regression
---
1: **Input:** Private data $\{(x_i, y_i)\}_{i=1}^n \in \left(\mathbb{R}^d \times \mathbb{R}\right)^n$. Predefined parameters $r, \tau_x, \tau_y, \lambda_n$.
2: **for** each user $i \in [n]$ **do**
3:     Clip $\bar{x}_i = x_i \min\{1, r/\|x_i\|_2\}$, i.e. $\bar{x}_i = \Pi_r(x_i)$. Release $\bar{x}_i \bar{x}_i^T$ to the server.
4:     Clip $\tilde{x}_i := \text{sgn}(x_i) \min\{|x_i|, \tau_x\}$ and $\tilde{y}_i := \text{sgn}(y_i) \min\{|y_i|, \tau_y\}$. Release $\tilde{x}_i \tilde{y}_i$ to the server.
5: **end for**
6: The server aggregates $\hat{\Sigma}_{\bar{X}\bar{X}} = \frac{1}{n} \sum_{i=1}^n \bar{x}_i \bar{x}_i^T$ and $\hat{\Sigma}_{\widetilde{X}\widetilde{Y}} = \frac{1}{n} \sum_{i=1}^n \tilde{x}_i \tilde{y}_i$.
7: Add noise $\dot{\Sigma}_{\bar{X}\bar{X}} = \hat{\Sigma}_{\bar{X}\bar{X}} + N_1$, where $N_1 \in \mathbb{R}^{d \times d}$ is a symmetric matrix and each entry of the upper triangular matrix is sampled from $\mathcal{N}(0, \frac{32 r^4 \log \frac{2.5}{\delta}}{n^2 \varepsilon^2})$. Add noise $\dot{\Sigma}_{\widetilde{X}\widetilde{Y}} = \hat{\Sigma}_{\widetilde{X}\widetilde{Y}} + N_2$, where the vector $N_2 \in \mathbb{R}^d$ is sampled from $\mathcal{N}(0, \frac{32 \tau_x^2 \tau_y^2 \log \frac{2.5}{\delta}}{n^2 \varepsilon^2} I_d)$.
8: Apply hard-thresholding to $\dot{\Sigma}_{\bar{X}\bar{X}}$ and obtain $\ddot{\Sigma}_{\bar{X}\bar{X}}$ where each entry is defined as $\ddot{\sigma}_{\bar{x}\bar{x}^T, ij} = \dot{\sigma}_{\bar{x}\bar{x}^T, ij} \cdot I[|\dot{\sigma}_{\bar{x}\bar{x}^T, ij}| > Thres]$, where $Thres = \gamma \sqrt{\frac{\log d}{n}} + \frac{4 r^2 \sqrt{2 \ln 1.25/\delta} \sqrt{\log d}}{n \varepsilon}$ and $\gamma$ is some constant.
9: The server outputs $\hat{\theta}^P(D) = S_{\lambda_n}([\ddot{\Sigma}_{\bar{X}\bar{X}}]^{-1} \dot{\Sigma}_{\widetilde{X}\widetilde{Y}})$.
---

**Lemma 1.** *The estimation error of private estimator $\ddot{\Sigma}_{\bar{X}\bar{X}}$ satisfies with probability $1 - d^{-\Omega(1)}$ that*

$$\mathbb{E}\|\ddot{\Sigma}_{\bar{X}\bar{X}} - \Sigma\|_\infty^2 \leq O\left(s^2 \left(\frac{\log d \log \frac{1}{\delta} r^4}{n \varepsilon^2}\right)^{1-q} + \left(\frac{\log d \log \frac{1}{\delta} r^4}{n \varepsilon^2}\right)\right)$$

*where the expectation takes over the randomness of the data records and the algorithm.*

*Remark* 1. When $q = 0$ and $r = O(1)$, Lemma 1 could achieve an error bound of $O(\frac{s^2 \log d}{n \epsilon^2})$, which matches the optimal rate of locally differentially private sparse covariance matrix estimation [51, 50].

Our private estimator is of the form $\hat{\theta}^P(D) = S_{\lambda_n}(\ddot{\Sigma}_{\bar{X}\bar{X}}^{-1} \dot{\Sigma}_{\widetilde{X}\widetilde{Y}})$. With some $r, \tau_x,$ and $\tau_y, \hat{\theta}^P(D)$, upper bound of $\tilde{O}(\frac{\sqrt{k}}{\sqrt{n}\varepsilon})$ is achievable. See Algorithm 1 for details. Notably, step 8 is the hard thresholding step for the private covariance matrix estimator. Importantly, the threshold only depends on $n, \log d$, and the privacy parameters and is independent on the two sparsities $s$ and $k$ of our problem. Moreover, since we assume $\|\theta^*\|_2 \leq \tau_\theta$, we can also project $\hat{\theta}^P(D)$ in Algorithm 1 onto a $\ell_2$-norm ball: $\bar{\theta}^P(D) = \Pi_{\tau_\theta}(\hat{\theta}^P(D))$, where $\Pi_{\tau_\theta}(v) = \arg\min_{v' \in \mathbb{B}(\tau_\theta)} \|v' - v\|_2^2$ and $\mathbb{B}(\tau_\theta)$ is the closed $\ell_2$-norm ball with radius $\tau_\theta$.

To sum up, high dimensionality gives rise to several consequences: (1) the regularization techniques used by [37, 14] are not applicable, (2) the invertibility of covariance matrix estimate is not guaranteed, (3) it also precludes the output perturbation method. Our proposed estimator properly overcomes the above challenges. To tackle (1) we embed the OLS estimator within the $\ell_1$ constraint to exploit the sparsity, along with a novel estimator of the covariance matrix to mitigate (2), and privatize it by sufficient perturbation to solve (3), which in turn makes our truthful analysis part much more complicated.

### 3.2 Payment Rule

We now turn our attention to the payment rule. The analyst wants to pay agents according to the veracity of the data they have provided and needs a reference against which to compare each data item. As mentioned before, the response is unverifiable, hence we lack a ground truth reference to validate the accuracy of the reported values. To circumvent the problem, we adopt the *peer prediction method* to determine a player's payment. In principle, the higher payment means higher consistency between the agent's reported value $\hat{y}_i$ and the predicted value of $y_i$ estimated using the collective input from their peers $\hat{D}_{-i}$.

To quantitatively measure the similarity between each agent's report and their peer's reports, we will use the *rescaled Brier score* rule[21]. Let $a_1$ and $a_2$ be positive parameters to be specified. Consider that $q$ represents the prediction of agent $i$'s response based on her own reports, and $p$ represents the prediction of agent $i$'s response based on her feature vector and her peers' reports. The analyst uses

**Algorithm 2** General Framework for Truthful and Private High Dimensional Linear Regression

1: Ask all agents to report their data $\hat{D}_1, \cdots, \hat{D}_n$;
2: Randomly partition agents into two groups, with respective data pairs $\hat{D}^0, \hat{D}^1$
3: For each dataset $\hat{D}, \hat{D}^0$ and $\hat{D}^1$, compute private estimators $\hat{\theta}^P(\hat{D})$, $\hat{\theta}^P(\hat{D}^b)$ for $b = 0, 1$ according to Algorithm 1
4: Compute estimators $\bar{\theta}^P(\hat{D}) = \Pi_{\tau_\theta}(\hat{\theta}^P(\hat{D}))$ and $\bar{\theta}^P(\hat{D}^b) = \Pi_{\tau_\theta}(\hat{\theta}^P(\hat{D}^b))$ for $b = 0, 1$
5: Set parameters $a_1, a_2$, and compute payments to each agent $i$: if agent $i$'s is in group $1 - b$, then he will receive payment $\pi_i = B_{a_1,a_2}\left(\langle \bar{x}_i, \bar{\theta}^P(\hat{D}^b)\rangle, \langle \bar{x}_i, \mathbb{E}_{\theta \sim p(\theta|\hat{D}_i)}[\theta]\rangle\right)$.

the following payment rule:

$$B_{a_1,a_2}(p, q) = a_1 - a_2(p - 2pq + q^2), \tag{3}$$

Observing that $B_{a_1,a_2}(p, q)$ is a function that exhibits strict concavity with respect to $q$, we point out that its maximum is attained when $q$ equals $p$. This implies a congruence between the prediction of agent $i$'s response based on her own information and that derived from her peers' information. For agent $i$, given $\mathbb{E}[y_i|x_i, \theta^*] = \langle x_i, \theta^*\rangle$, it is logical to set $p = \langle \bar{x}_i, \bar{\theta}^P(\hat{D}^b)\rangle$ and $q = \langle \bar{x}_i, \mathbb{E}_{\theta \sim p(\theta|\hat{D}_i)}[\theta]\rangle$. Here, $\bar{\theta}^P(\hat{D}^b)$ denotes the private estimator for a dataset $\hat{D}^b$ excluding $\hat{D}_i$, and $p(\theta|\hat{D}_i)$ represents the posterior distribution of $\theta$ post the analyst's receipt of $\hat{D}_i$.

Building upon this analysis, we structure our Algorithm 2. This algorithm utilizes reported data, which may contain manipulated responses, to generate estimators. To maintain data independence, we divide the dataset into two subsets, $\hat{D}^0$ and $\hat{D}^1$. For the purpose of calculating the payment for each agent $i$ in group $b \in \{0, 1\}$, the estimator $\theta^*$ is derived using $\hat{D}^{1-b}$. Finally, the algorithm applies $\bar{\theta}^P(\hat{D})$ in combination with the specific agent's feature vector $x_i$ to forecast their response.

## 4 Theoretical Results and Implementation
### 4.1 Accuracy and Privacy Analysis

**Theorem 2** (Privacy). *The output of Algorithm 2 satisfies* $(2\varepsilon, 3\delta)$*-JDP.*

*Remark* 2. By the selection of $\varepsilon$ and $\delta$, our mechanism could achieve an $(o(\frac{1}{\sqrt{n}}), O(n^{-\Omega(1)}))$-JDP, which provides asymptotically the same good privacy guarantee as in [14] for the low dimension setting with bounded covariates and responses. Specifically, [14] shows that it is possible to design an $o(\frac{1}{\sqrt{n}})$-JDP mechanism, while here we consider the approximate JDP due to the Gaussian noise we add. [37] considers truthful (low dimensional) linear regression with sub-Gaussian/heavy-tailed covariates and responses, our Theorem 2 is better than theirs. In detail, [37] can only guarantee Random Joint Differential Privacy (RJDP) where on all but a small proportion of unlikely dataset pairs, pure $\varepsilon$-JDP holds, while in this paper we can guarantee approximate JDP, which is more widely used in the DP literature. The main reason is that [37] used the output perturbation-based method to ensure DP. However, as the data distribution is sub-Gaussian rather than bounded as in [14], the sensitivity of the closed-form linear regression estimator could be extremely large (with some probability). Thus, the output perturbation-based method can fail with a small probability and can only ensure RJDP. Instead, in our method, we use sufficient statistics perturbation, due to our clipping operator, the sensitivities of $\hat{\Sigma}_{\bar{X}\bar{X}}$ and $\hat{\Sigma}_{\widetilde{X}\widetilde{Y}}$ are always bounded. One consequence of adopting our sufficient statistics perturbation method is that the utility analysis is harder than that of the output perturbation-based method, as we will discuss in the following.

**Theorem 3** (Accuracy). *Fix a privacy parameter* $\varepsilon$*, a participation goal* $1 - \alpha$ *and a desired confidence parameter* $\beta$ *in Definition 6. Then under the symmetric threshold strategy* $\sigma_{\tau_{\alpha,\beta}}$*, when $n$ is sufficient large such that* $n \geq \Omega(\frac{s^2 r^4 \log d \log \frac{1}{\delta}}{\varepsilon^2 \kappa_\infty})$*, the output* $\bar{\theta}^P(\hat{D})$ *of Algorithm 2 satisfies that with probability at least* $1 - \beta - O(n^{-\Omega(1)})$*,*

$$\mathbb{E}[\|\bar{\theta}^P(\hat{D}) - \theta^*\|_2^2] \leq O\left(\left(\alpha^2 n + \frac{1}{n}\right)\frac{kr^4 \log d \log \frac{1}{\delta}}{\varepsilon^2}\right).$$

*Remark* 3. The above bound is independent on $\text{Poly}(d)$. This outcome is primarily due to the use of sufficient statistics perturbation method, which effectively reduces the size of noise. It is notable that

Assumption 6 only ensures that when $n \geq \tilde{\Omega}(\frac{s^2 r^4}{\varepsilon^2 \kappa_\infty})$ our private covariance estimator is invertible. Thus, even if the assumption does not hold, we can still get the same upper bound as in Theorem 3 as long as $n \geq \tilde{\Omega}(d)$.

Our framework for analyzing the accuracy differs dramatically from that of the prior ones since our privatization mechanism incurs finer and more delicate analysis on some specific error terms. The work of [37] and [14] both used the output perturbation method, as it provides an explicit characterization of the noise. Specifically, their estimator can be represented as $\bar{\theta}^P(\hat{D}) = \Pi_{\tau_\theta}(\tilde{\theta}^P(\hat{D}))$, where $\tilde{\theta}^P(\hat{D}) = \tilde{\theta}(\hat{D}) + v$ with $\tilde{\theta}(\hat{D})$ as a (non-private) closed-form estimator and $v$ is the added Gamma noise of scale $O(\sqrt{d/n})$ to the non-private estimator. It can be shown that:

$$\mathbb{E}[\|\bar{\theta}^P(\hat{D}) - \theta^*\|_2^2] \leq \mathbb{E}\|\tilde{\theta}^P(\hat{D}) - \theta^*\|_2^2 \leq 2\mathbb{E}\|\tilde{\theta}^P(\hat{D}) - \hat{\theta}(D)\|_2^2 + 2\mathbb{E}\|\hat{\theta}(D) - \theta^*\|_2^2$$
$$= 2(\mathbb{E}\|\tilde{\theta}(\hat{D}) - \hat{\theta}(D)\|_2^2 + \mathbb{E}\|v\|_2^2 + \mathbb{E}\|\hat{\theta}(D) - \theta^*\|_2^2) \tag{4}$$

where $D$ is original data and $\hat{D}$ is the reported data. Note that the second and the third terms in (4) are easy to upper bound. For the first term, since we know the number of manipulated data in $\hat{D}$ is bounded by $\alpha n$ with high probability, indicating there are at most $\alpha n$ different samples between $\hat{D}$ and $D$. Thus, it can be bounded by directly re-using the sensitivity analysis of $\tilde{\theta}(D)$ as a part of the privacy data analysis. However, as we mentioned previously, the sensitivity of (2) is of scale $O(\sqrt{d/n})$, indicating the previous method cannot be applied in our setting. Hence our tactic is to privatize the terms $\hat{\Sigma}_{\bar{X}\bar{X}}$ and $\hat{\Sigma}_{\tilde{X}\tilde{Y}}$ separately. We shall take another route to estimate the error of the private estimator $\bar{\theta}^P(\hat{D})$:

$$\mathbb{E}[\|\bar{\theta}^P(\hat{D}) - \theta^*\|_2^2] \leq \mathbb{E}\|\hat{\theta}^P(\hat{D}) - \theta^*\|_2^2 \leq 2\mathbb{E}\|\hat{\theta}^P(\hat{D}) - \hat{\theta}^P(D)\|_2^2 + 2\mathbb{E}\|\hat{\theta}^P(D) - \theta^*\|_2^2.$$

The above framework is now estimating the difference in the private $\hat{\theta}^P$ (instead of the non-private $\hat{\theta}$). However, this nuance inevitably leads to more complex analysis, i.e. the term $\mathbb{E}\|\hat{\theta}^P(\hat{D}) - \hat{\theta}^P(D)\|_2^2$ is much more complicated to deal with than the first term in (4). The bound on $\mathbb{E}\|\hat{\theta}^P(\hat{D}) - \hat{\theta}^P(D)\|_2^2$ cannot be obtained by simply applying existing results on the covariance matrix estimation as in [37] or the assumption of strong convexity of the loss function as in [14]. Specifically, we tackle this issue by combining the analysis for $s$-approximately sparse covariance matrices and some technical tools such as the decomposability of the $\ell_1$ norm term in (1) to give a non-trivial bound. The relevant lemma is given below and as mentioned this bound comprises a key component for the proof of Theorem 3.

**Lemma 4.** *Let $\hat{\theta}^P(\hat{D})$ and $\hat{\theta}^P(D)$ be the private estimators on the original dataset $D$ and the reported dataset $\hat{D}$ that at most one agent misreports. We set the constraint bound $\lambda_n = O\left(\frac{r^2 \sqrt{\log d \log \frac{1}{\delta}}}{\sqrt{n}\varepsilon}\right)$, when n is sufficient large such that $n \geq \Omega(\frac{s^2 r^4 \log d \log \frac{1}{\delta}}{\varepsilon^2 \kappa_\infty})$, with probability at least $1 - O(d^{-\Omega(1)})$ we have the following error bounds:*

$$\|\hat{\theta}^P(\hat{D}) - \hat{\theta}^P(D)\|_\infty \leq 4\lambda_n, \left\|\hat{\theta}^P(\hat{D}) - \hat{\theta}^P(D)\right\|_2 \leq 16\sqrt{k}\lambda_n.$$

## 4.2 Analysis for the Properties of Truthful Mechanism

**Theorem 5** (Truthfulness). *Fix a privacy parameter $\varepsilon$, a participation goal $1 - \alpha$ and a desired confidence parameter $\beta$ in Definition 6. Then with probability at least $1 - \beta - O(n^{-\Omega(1)}) - nd^{-\frac{9}{2}}$, the symmetric threshold strategy $\sigma_{\tau_{\alpha,\beta}}$ is an $\eta$-approximate Bayesian Nash equilibrium in Algorithm 2 with*

$$\eta = O\left(a_2 r^4 \log d \log \frac{1}{\delta} \left(\frac{n\alpha^2 k}{\varepsilon^2} + \frac{1}{n\varepsilon^2}\right) + \tau_{\alpha,\beta}\delta\varepsilon^3\right).$$

We can see with some specified parameters, the above bound could be sufficiently small. For example, when we set $\alpha = O(\frac{1}{n})$ we have $\eta \to 0$ as $n \to \infty$ since $\delta = o(\frac{1}{n})$.

**Theorem 6** (Individual rationality). *With probability at least $1 - \beta - O(n^{-\Omega(1)}) - nd^{-\frac{9}{2}}$, Algorithm 2 is individually rational for all agents with cost coefficients $c_i \leq \tau_{\alpha,\beta}$ as long as*

$$a_1 \geq a_2(r\tau_\theta + 3r^2 \tau_\theta^2) + \tau_{\alpha,\beta}8(1 + 3\delta)\varepsilon^3$$

*regardless of the reports from agents with cost coefficients above $\tau_{\alpha,\beta}$, where $a_1, a_2$ are in (6).*

*Remark* 4. Our mechanism may not be individually rational for all agents, since the privacy cost coefficients follow an unbounded distribution with exponential decay. This assumption is made reasonable because for a relatively small subset of agents the privacy costs could be exceptionally high, indicating certain individuals may persistently refrain from reporting truthfully, regardless of the compensation offered to them. The same situation happens in other truthful mechanisms as well.

**Theorem 7** (Budget). *With probability at least $1 - \beta - O(n^{-\Omega(1)}) - nd^{-\frac{9}{2}}$, the total expected budget $\mathcal{B} = \sum_{i=1}^{n} \mathbb{E}(\pi_i)$ required by the analyst to run Mechanism 1 under threshold equilibrium strategy $\sigma_{\tau_{\alpha,\beta}}$ satisfies*

$$\mathcal{B} \le n(a_1 + a_2(r\tau_\theta + r^2\tau_\theta^2)).$$

*Remark* 5. With some appropriate setting of $\alpha$, it is reasonable to expect the overall expected budget to diminish toward zero as the sample size increases. This is because if the sample size is infinite, the ground-truth parameters can always recovered by our estimator, which allows the analyst to pay nothing to incentivize the agents.

## 4.3 Formal Statement of Main Result

**Corollary 8.** *For any $\xi \in (\frac{1}{3}, \frac{1}{2})$ and $c > 0$, we set $\varepsilon = n^{-\xi}$, $\delta = \Theta(n^{-\Omega(1)})$, $\alpha = \Theta(n^{-3\xi})$, $\beta = \Theta(n^{-c})$, $a_2 = O(n^{-3\xi})$, $a_1 = a_2(r\tau_\theta + 3r^2\tau_\theta^2) + \tau_{\alpha,\beta}8(1 + 3\delta)\varepsilon^3$. Then the output of Algorithm 2 satisfies $(O(n^{-\xi}), O(n^{-\Omega(1)}))$-JDP. Moreover, with probability at least $1 - O(n^{-\Omega(1)})$, it holds that:*

1. *The symmetric threshold strategy $\sigma_{\tau_{\alpha,\beta}}$ is a $\widetilde{O}(n^{-\xi-1})$-Bayesian Nash equilibrium for a $1 - O(n^{-3\xi})$ fraction of agents to truthfully report their data;*

2. *The private estimator $\bar{\theta}^P(\hat{D})$ is $\widetilde{O}(n^{2\xi-1})$-accurate;*

3. *It is individually rational for a $1 - O(n^{-3\xi})$ fraction of agents to take part in the mechanism;*

4. *The total expected budget required by the analyst is $\widetilde{O}(n^{1-3\xi})$.*

*Remark* 6. It is important to highlight the subtle balance between the precision of the estimator and the other attributes of the mechanism. Compromising on accuracy often results in several consequences: (1) a reduction in the total compensation paid to the agents, (2) an increase in the proportion of rational agents, and (3) a closer approximation to the Bayesian Nash Equilibrium. However, we do not observe such a trade-off in [37]'s implementation for linear regression cases. Besides, our results slightly differ from that of [14] in the sense that they do not have a trade-off between rationality and accuracy. Such discrepancies may be caused by varying willingness to supply higher compensation to the agents, resulting in different settings of parameters.

## 5 Conclusions

In this paper, we fit the high dimensional sparse linear regression privately and we propose a truthful mechanism to incentivize the strategic users. On the one hand, the mechanism is $(o(1), O(n^{-\Omega(1)}))$-jointly differentially private. This is achieved by using the sufficient statistics perturbation method which adds much less noise than the output perturbation method employed by prior work. Leveraging the idea of *soft-thresholding*, we propose a private estimator of $\theta^*$ to exploit the sparsity of the model and it achieves an error of $o(1)$. On the other hand, via some computation on the consistency between the agent's reported value and the predicted value using peers' reports, we design an appropriate payment scheme for each agent using *rescaled Brier score* rule. This method ensures that our mechanism reaches an $o(\frac{1}{n})$-approximate Bayes Nash equilibrium for a $(1 - o(1))$-fraction of agents to truthfully report their data while a $(1 - o(1))$ fraction of agents receive non-negative utilities, thus establishing their individual rationality. Moreover, the total payment budget required from the analyst to run the mechanism is $o(1)$.

## Acknowledgments

Di Wang and Liyang Zhu were supported in part by the funding BAS/1/1689-01-01, URF/1/4663-01-01, REI/1/5232-01-01, REI/1/5332-01-01, and URF/1/5508-01-01 from KAUST, and funding from KAUST - Center of Excellence for Generative AI, under award number 5940. Jinyan Liu is supported by the National Natural Science Foundation of China (NSFC Grant No. 62102026).

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

# A Notations and Omitted Relevant Definitions

| Notation | Description |
|---|---|
| $n$ | The number of agents |
| $d$ | The dimensionality |
| $x_i \in \mathcal{X} \subseteq \mathbb{R}^d$ | The feature vector of agent $i$ |
| $y_i \in \mathcal{Y} \subseteq \mathbb{R}$ | The response variable of agent $i$ |
| $\Sigma_{XX}$ or $\Sigma$ | The covariance matrix of $x_i$ |
| $\|X\|_p$ | $p$-norm of matrix $X$ |
| $I[A]$ | Indicator of event $A$ |
| $\mathrm{sgn}(x)$ | Sign function of a real number $x$ |
| $[S_\lambda(u)]_i$ | Element-wise soft-thresholding operator |
| $N_1 \in \mathbb{R}^{d \times d}$ | Gaussian noise added to $\hat{\Sigma}_{\bar{X}\bar{X}}$ |
| $N_2 \in \mathbb{R}^d$ | Gaussian noise added to $\hat{\Sigma}_{\widetilde{X}\widetilde{Y}}$ |
| $\gamma$ | Some constant associated with the variance of $N_2$ |
| $\widetilde{y}_i$ | shrunken version of $y_i$ via parameter $\tau_y$ |
| $\hat{\Sigma}_{\bar{X}\bar{X}}(\hat{\Sigma}_{\widetilde{X}\widetilde{Y}})$ | The sample covariance between $X$ and $X$ (The sample covariance between the clipped $X$ and the clipped $Y$) |
| $\dot{\Sigma}_{\bar{X}\bar{X}}(\dot{\Sigma}_{\widetilde{X}\widetilde{Y}})$ | a noisy and (clipped) version of $\hat{\Sigma}_{\bar{X}\bar{X}}$ ($\hat{\Sigma}_{\widetilde{X}\widetilde{Y}}$) |
| $c_i \in \mathcal{C} \subseteq \mathbb{R}_+$ | The privacy cost coefficient of agent $i$ |
| $\mathcal{D} = \mathcal{X} \times \mathcal{Y}$ | The feature-response data universe |
| $D_i = (x_i, y_i)$ | The feature-response data pair of agent i |
| $\sigma_i(\cdot, \cdot) : \mathcal{D} \times \mathcal{C} \to \mathcal{Y}$ | The reporting strategy of agent i |
| $\hat{y}_i = \sigma_i(D_i, c_i)$ | The reported response of agent i |
| $\hat{D}_i = (x_i, \hat{y}_i)$ | The reported feature-response data pair of agent $i$ |
| $D = \{D_i\}_{i=1}^n$ | All feature-response data pairs |
| $\hat{D} = \left\{\hat{D}_i\right\}_{i=1}^n$ | All reported feature-response data pairs |
| $\sigma = \{\sigma_i\}_{i=1}^n$ | The collection of all agents' reporting strategies |
| $D_{-i} = D \backslash D_i$ | The collection of data pairs except $D_i$ |
| $\hat{D}_{-i} = \hat{D} \backslash \hat{D}_i$ | The collection of reported data pairs except $\hat{D}_i$ |
| $\sigma_{-i} = \sigma \backslash \sigma_i$ | The collection of strategies except $\sigma_i$ |
| $c_{-i} = \{c_j\}_{j=1}^n \backslash c_i$ | The collection of privacy cost coefficients except $c_i$ |
| $\pi_i$ | The payment to agent $i$ |
| $u_i$ | The utility of agent $i$ |
| $\varepsilon, \delta \in \mathbb{R}_+$ | The privacy parameters of the mechanism |
| $\theta^* \in \mathbb{R}^d$ | The model parameter needs to estimate |
| $\sigma_\tau$ | The threshold strategy with privacy cost threshold $\tau \in \mathbb{R}_+$ |
| $\tau_{\alpha,\beta}$ | The privacy cost threshold characterized by $\alpha, \beta \in (0, 1)$ |
| $\mathcal{B}$ | The expected total budget |
| $\hat{\theta}(D)$ | The non-private estimator on data $D$ |
| $\hat{\theta}^P(\hat{D})$ | The private estimator on the reported data $\hat{D}$ |
| $\bar{\theta}^P(\hat{D})$ | The private estimator on the reported data $\hat{D}$ after projection |
| $\kappa_2, \kappa_\infty$ | The lower bounds for the operator norm and infinity norm of matrix $\Sigma$ respectively |
| $\eta$ | The level of truthfulness |
| $\xi \in (\frac{1}{3}, \frac{1}{2})$ | The parameter to control the trade-off between accuracy and truthfulness, payment amount and individual rationality. |

## A.1 Definitions of Differential Privacy

**Definition 1** (Differential Privacy [17]). *Given a data universe $\mathcal{X}$, we say that two datasets $D, D' \subseteq \mathcal{X}$ are neighbors if they differ by only one entry, which is denoted as $D \sim D'$. A randomized algorithm $\mathcal{A}$ is $(\varepsilon, \delta)$-differentially private (DP) if for all neighboring datasets $D, D'$ and for all events $S$ in the output space of $\mathcal{A}$, we have $\mathbb{P}(\mathcal{A}(D) \in S) \leq e^\varepsilon \mathbb{P}(\mathcal{A}(D') \in S) + \delta$.*

**Definition 2.** *(Gaussian Mechanism). Given any function $q : \mathcal{X}^n \to \mathbb{R}^p$, the Gaussian Mechanism is defined as: $\mathcal{M}_G(D, q, \varepsilon) = q(D) + Y$, where $Y$ is drawn from Gaussian Distribution $\mathcal{N}\left(0, \sigma^2 I_p\right)$ with $\sigma \geq \sqrt{2 \ln(1.25/\delta)} \Delta_2(q)/\varepsilon$. Here $\Delta_2(q)$ is the $\ell_2$-sensitivity of the function $q$, i.e. $\Delta_2(q) = \sup_{D \sim D'} \| q(D) - q(D') \|_2$. Gaussian Mechanism preserves $(\varepsilon, \delta)$-differential privacy.*

**Definition 3** (Joint Differential Privacy [31]). *Consider a randomized mechanism $\mathcal{M} : \mathcal{D}^n \to \Theta \times \Pi^n$ with arbitrary response sets $\Theta, \Pi^n$. For each $i \in [n]$, we denote by $\mathcal{M}(\cdot)_{-i} = (\theta, \pi_{-i}) \in \Theta \times \Pi^{n-1}$ the portion of the mechanism's output that is observable by external observers and agents $j \neq i$. a mechanism $\mathcal{M}$ preserves $(\varepsilon, \delta)$- joint differential privacy (JDP), at privacy level $\varepsilon > 0$ and confidence level $\delta \in (0, 1)$, if for every agent $i$, every dataset $D \in \mathcal{D}^n$ and every $D_i, D_i' \in \mathcal{D}$ we have $\forall \mathcal{S} \subseteq \Theta \times \Pi^{n-1}$,*

$$\mathbb{P}\left(\mathcal{M}(D_i, D_{-i})_{-i} \in \mathcal{S} | (D_i, D_{-i})\right) \leq$$
$$e^\varepsilon \mathbb{P}\left(\mathcal{M}(D_i', D_{-i})_{-i} \in \mathcal{S} | (D_i', D_{-i})\right) + \delta,$$

*where $D_{-i} \in \mathcal{D}^{n-1}$ is the dataset $D$ that excludes the $i$-th sample in $D$ and $\pi_{-i}$ is the vector that contains all payments except the payment of agent $i$.*

An $(\varepsilon, \delta)$-JDP mechanism on any dataset (including likely ones) may leak sensitive information on low probability responses, forgiven by the additive $\delta$ relaxation, which we hope to have a magnitude of $o(\frac{1}{n})$. Definition 3 assumes that the private estimator $\bar{\theta}$ produced by the mechanism $\mathcal{M}$ is publicly observable, and that the payments $\pi_i$ can only be seen by agent $i$. Thus, from the perspective of each agent $i$, the released public output $(\theta, \pi_{-i})$ may compromise their privacy.

## A.2 Truthful mechanism properties

Consider $\mathcal{M}$ the regression mechanism that takes as input the reported data $\hat{D} = \{\hat{D}_i = (X_i, \hat{y}_i)\}_{i=1}^n$ from n agents, with $\hat{y}_i = \sigma_i(D_i)$ the reporting strategy adopted by agent $i$. We denote the set of all agents' strategies by the strategy profile $\sigma = (\sigma_1, ..., \sigma_n)$ and $\sigma_{-i} = (\sigma_1, ..., \sigma_{i-1}, \sigma_{i+1}, ..., \sigma_n)$ denotes the set of strategies excluding the reporting strategy of agent $i$. Let $f_i(c_i, \varepsilon, \delta)$ be the privacy cost function of agent $i$ with $c_i$ her cost parameter. Each agent $i$ receives a real-valued payment $\pi_i$ and finally receives utility $u_i = \pi_i - f_i(c_i, \varepsilon, \delta)$. Based on these settings, we quantify property (1) by introducing Bayesian Nash equilibrium.

**Definition 4** ($\eta$-Bayesian Nash Equilibrium). *A strategy profile $\sigma = (\sigma_1, ..., \sigma_n)$ forms an $\eta$-Bayesian Nash Equilibrium if for every agent $i$, $D_i$ and $c_i$, and for every reporting strategy $\sigma_i' \neq \sigma_i$, one have :*

$$\mathbb{E}_{D_{-i}, c_{-i} \sim p(D_{-i}, c_{-i} | D_i, c_i)}[u_i(\sigma_i(D_i, c_i), \sigma_{-i}(D_{-i}, c_{-i}))] \geq$$
$$\mathbb{E}_{D_{-i}, c_{-i} \sim p(D_{-i}, c_{-i} | D_i, c_i)}[u_i(\sigma_i'(D_i, c_i), \sigma_{-i}(D_{-i}, c_{-i}))] - \eta.$$

The positive value $\eta$ represents the maximum of additional expected payment an agent might receive by altering their reporting strategy. To encourage agents to report their data truthfully, the payment rule should aim to keep $\eta$ as minimal as possible. This paper investigates a specific threshold strategy. We will establish that if all agents consistently employ the threshold strategy with a common positive value $\tau$, this unified strategy profile will effectively realize an $\eta$-Bayesian Nash equilibrium.

**Definition 5** (Threshold Strategy). *The threshold strategy $\sigma_\tau$ is defined as follows:*

$$\hat{y}_i = \sigma_\tau(x_i, y_i, c_i) = \begin{cases} y_i, & \text{if} \quad c_i \leq \tau, \\ \text{arbitrary value in } \mathcal{Y}, & \text{if} \quad c_i > \tau. \end{cases}$$

**Definition 6** (Threshold $\tau_{\alpha, \beta}$). *Fix a probability density function $p(c)$ of privacy cost parameter, and let*

$$\tau_{\alpha,\beta}^1 = \inf\{\tau > 0 : \mathbb{P}_{(c_1, \cdots, c_n) \sim p^n}(\#\{i : c_i \leq \tau\} \geq (1 - \alpha)n) \geq 1 - \beta\},$$
$$\tau_\alpha^2 = \inf\{\tau > 0 : \inf_{D_i} \mathbb{P}_{c_j \sim p(c|D_i)}(c_j \leq \tau) \geq 1 - \alpha\}.$$

*Define $\tau_{\alpha,\beta}$ as the larger of these two thresholds: $\tau_{\alpha,\beta} = \max\{\tau_{\alpha,\beta}^1, \tau_\alpha^2\}$.*

$\tau_{\alpha,\beta}^1$ is a threshold that with probability at least $1 - \beta$, at least $1 - \alpha$ fraction of agents have cost coefficient $c_i \leq \tau_{\alpha,\beta}$. $\tau_\alpha^2$ is a threshold that conditioned on their own dataset $D_i$, each agent $i$ believes that with probability $1 - \alpha$ any other agent $j$ has cost coefficient $c_j \leq \tau_{\alpha,\beta}$.

We introduce the definition of $\eta$-accuracy and we use the $l_2$-norm distance between the private estimator and the true parameter.

**Definition 7** ($\eta$-accuracy). *A mechanism is $\eta$-accurate if its output $\bar{\theta}^P$ satisfies $\mathbb{E}\left[\|\bar{\theta}^P - \theta^*\|_2^2\right] \leq \eta$.*

**Definition 8** (Individual Rationality). *A mechanism is individually rational if the utility received by each agent $i$ has a non-negative expectation: $\forall i \in [n], \mathbb{E}[u_i] \geq 0$*

To satisfy individual rationality, the mechanism should output payments high enough to compensate for privacy costs. Meanwhile, we also expect a total payment budget $\mathcal{B} = \sum_{i=1}^{n} \mathbb{E}[\pi_i] = o(1)$ that tends to zero as the number of agents $n$ increases.

**Definition 9** (Asymptotically Small Budget). *An asymptotically small budget is such that $\mathcal{B} = \sum_{i=1}^{n} \mathbb{E}[\pi_i] = o(1)$ for all realizable dataset $D = \{(x_i, y_i)\}_{i=1}^n$.*

To sum up, we have $n$ agents reporting potentially manipulated data, denoted as $\hat{D} = (x_i, y_i)$, to an analyst. Subsequently, the analyst computes an estimator for the model parameters using the reported data. To preserve DP, the analyst meticulously introduces controlled noise into the estimator. Furthermore, the analyst strategically compensates the agents for their privacy costs based on their reports, with the goal of incentivizing truthful reporting. This incentivization is implemented under the assumption that the agents will adhere to a threshold strategy.

# B   Related Work

Following the pioneering work of [22], a substantial body of research has explored data acquisition problems involving agents with privacy concerns [20, 32, 57, 15, 21, 18, 19, 13]. The majority of this work operates in a model where agents cannot fabricate their private information; their only recourse is either to withhold it or, at most, misrepresent their costs related to privacy. A related thread [22, 34, 10] explores cost models based on the notion of differential privacy [17]. However, most of the aforementioned work only considers the mean estimation problem and does not consider advanced statistical models [18, 19, 13]. Thus, these methods cannot be applied to linear models. [14] and [37] represent the closest work to ours, as they investigate the challenge of estimating (generalized) linear models from self-interested agents that have privacy concerns. However, as we mentioned above, they only consider the low dimensional case and their methods cannot be applied to the high dimensional sparse setting, see Section C for details.

Classical approaches including the work by [55, 27, 43, 44, 45, 58, 1, 30, 41, 8, 3, 52, 54] for estimating linear regression models with differential privacy (DP) guarantee utilize central and local models. These approaches typically involve adding noise to the output of optimization methods, privatizing the objective function, or injecting noise into the gradients during optimization. However, the truthful data acquisition process follows a single-round interactive procedure between the analyst and the agents, in contrast to the multi-round interactions typically encountered in these approaches, indicating these methods cannot be applied to our problem. The closed-form expression of our estimator ensures that this requirement is satisfied. We believe that our method is non-trivial and it has the potential to extend to other related scenarios. Recently, [61] proposed a closed-form private estimator for sparse linear regression. However, as we discussed in Section 3.1, their estimator cannot be directly applied to our problem due to inherent limitations, such as challenges associated with the invertibility of the covariance matrix.

Recently, there has been also some worth-noting literature on estimating linear regression from strategic agents (without privacy consideration). [25] and [6] have investigated regressing a linear model using data from strategic agents who can manipulate their associated costs but not the data. [29] explored a setting without direct payments, in which agents receive a utility that serves as a measure of estimation accuracy. [11] studied the case where agents may intentionally introduce errors to maximize their own benefits and proposed several group-strategyproof linear regression mechanisms. Additionally, [16] and [35] considered an analyst who aims to derive a "consensus" model from data provided by multiple strategic agents. In this setting, agents seek a consensus value that minimizes their individual data loss, and the authors demonstrated that empirical risk minimization is group-strategyproof. However, all of these methods only consider the low dimensional setting and cannot be applied to the high dimensional sparse linear regression.

## C Challenges in Designing Truthful and Privacy-preserving Sparse Linear Regression

In this section, we provide a retrospect of previous methods and then outline the explain why existing methods are not suitable for directly applying to our settings. The challenges associated with the problem make it fundamentally difficult to design a desirable mechanism.

Under truthful setting, [14] proposed to use the general Ridge regression estimator, i.e., solving the regularized convex program: $\arg\min_\theta \frac{1}{2n}\|Y - X\theta\|_2^2 + \gamma\|\theta\|_2^2$ where $\gamma$ is some parameter and $X = (x_1^T, \cdots, x_n^T)^T \in \mathbb{R}^{n \times d}$, and $Y = (y_1, \cdots, y_n)^T \in \mathbb{R}^n$. The unique closed-form solution can be written as $\hat{\theta}^R(D) = (\gamma I + X^T X)^{-1} X^T Y$. [37] considered the classical ordinary least square (OLS) estimator $\hat{\theta}^{OLS} = (\hat{\Sigma}_{XX})^{-1}\hat{\Sigma}_{XY}$ where $\hat{\Sigma}_{XX} = \frac{1}{n}\sum_{i=1}^n x_i x_i^T$ and $\hat{\Sigma}_{XY} = \frac{1}{n}\sum_{i=1}^n x_i y_i$. Both of the private estimators are based on the closed-form solution $\hat{\theta}^L(D)$ of the ridge regression. However, there are four problems with the above methods, $\hat{\theta}^R$ and $\hat{\theta}^{OLS}$.

### C.1 Sparsity-encouraging and Private Regressor

The $\ell_2$-norm accuracy errors of $\hat{\theta}^R$ and $\hat{\theta}^{OLS}$ are both bounded by $O(\sqrt{d/n})$. This rate is not convergent in the data-poor scenario, indicating that both of these two estimators are not suitable in our setting. Since $\ell_2$-regularization used in $\hat{\theta}^R$ is not a sparsity-encouraging technique, we shall seek some alternative regularization. A qualified estimator that satisfies our need must (1) exploit the sparsity assumption and (2) have a closed-form solution.

Apart from the choice of (non-private) estimator, we also need to give some thoughts on the privacy mechanism. Previous work [14] and [37] both adopt the output perturbation method to ensure $\varepsilon$-JDP, i.e., adding some Gamma noise to $\hat{\theta}^R$ or $\hat{\theta}^{OLS}$. According to the DP theory, the scale of the noise should be proportional to the $\ell_2$-norm sensitivity, which can be readily bounded by $\sqrt{d/n}$ through clipping operation or if the data are assumed to be bounded. In the high dimensional setting, output perturbation could fail catastrophically as $d \gg n$. The magnitude of noise needs to grow polynomially with $\sqrt{d/n}$, causing the estimation error much larger. This means we must find a privatization mechanism that gives a better accuracy guarantee.

### C.2 The Invertibility of Covariance Matrix

The invertibility of the covariance matrix is also an issue in high dimensional space. The classical OLS estimator $\hat{\theta}^{OLS}$ used in [37] is no longer well-defined as $\hat{\Sigma}_{\bar{X}\bar{X}}$ is not full-rank. In the case of $\hat{\theta}^R = (\varrho I + X^T X)^{-1} X^T Y$, although $(\varrho I + X^T X)^{-1}$ always exists for some $\varrho > 0$, the caveat with such method is however its concentration only holds when $n = \Omega(d)$ [46].

To solve the invertibility issue posed by high dimensionality, we need to consider properly imposing some reasonable assumption on the structure of covariance matrix $\Sigma$, which will be formally introduced and serve to yield an efficient estimator for the covariance matrix with such assumption.

### C.3 Truthfulness Analysis Framework

Due to the additional regularization parameter $\varrho$, $\hat{\theta}^L$ is a biased estimator of $\theta^*$, which complicates the truthfulness analysis. This complexity arises because the OLS linear regression estimator $\hat{\theta}^{OLS} = (X^T X)^{-1} X^T y$ maximizes the scoring rule when players truthfully report their responses. However, for $\hat{\theta}^L(D)$, the bias caused by $\varrho$ causes the optimal report of each player to deviate from truthful reporting by a quantity proportional to the bias. To address the above issue, [14] shows that as long as $\varrho$ grows more slowly than $n$, the bias term of $\hat{\theta}^L$ converges to zero with high probability. More specifically, it is shown that the estimation error can be decomposed into the summation of a bias and a variance term: $\mathbb{E}[\|\hat{\theta}(D) - \theta^*\|_2^2] = \mathrm{trace}(\mathrm{Cov}(\hat{\theta}(D))) + \|\mathrm{bias}(\hat{\theta}(D))\|_2^2$, where $\mathrm{trace}(\mathrm{Cov}(\hat{\theta}(D))) = \sigma^2 (\varrho I + X^T X)^{-1} X^T X (\varrho I + X^T X)^{-1}$, $\mathrm{bias}(\hat{\theta}(D)) = -\varrho (\varrho I + X^T X)^{-1} \theta^*$, $\sigma^2$ is the variance of the noise variables $\zeta_i$. However, it remains uncertain whether there is a closed-form bias-variance decomposition available for our estimator proposed in Section 3.1.

# D  Limitation

Although some truthful mechanism studies [12] provided payment schemes for multiple-round data acquisition, existing truthful linear regression research poses constraint to the mechanism where data are only collected once. In this paper we mainly follow the analysis framework of [14] to guarantee truthful mechanism properties, therefore our work is still hampered by the one-round communication constraint. It remains uncertain whether this constraint could be removed in the future.

# E  Supporting lemmas

**Definition 10** (Sub-Gaussian random variable). *A zero-mean random variable $X \in \mathbb{R}$ is said to be sub-Gaussian with variance $\sigma^2$ $\left(X \sim \mathrm{subG}\left(\sigma^2\right)\right)$ if its moment generating function satisfies $\mathbb{E}[\exp(tX)] \leq \exp\left(\frac{\sigma^2 t^2}{2}\right)$ for all $t > 0$. For a sub-Gaussian random variable $X$, its sub-Gaussian norm $\|X\|_{\psi_2}$ is defined as $\|X\|_{\psi_2} = \inf\{c > 0 : \mathbb{E}[\exp(\frac{X^2}{c^2})] \leq 2\}$. Specifically, if $X \sim subG(\sigma^2)$ we have $\|X\|_{\psi_2} \leq O(\sigma)$.*

**Definition 11** (Sub-Gaussian random vector). *A zero mean random vector $X \in \mathbb{R}^d$ is said to be sub-Gaussian with variance $\sigma^2$ (for simplicity, we call it $\sigma^2$-sub-Gaussian), which is denoted as $\left(X \sim \mathrm{subG}_d\left(\sigma^2\right)\right)$, if $\langle X, u \rangle$ is sub-Gaussian with variance $\sigma^2$ for any unit vector $u \in \mathbb{R}^d$.*

**Lemma 9.** *For a sub-Gaussian vector $X \sim \mathrm{sub}\, G_d\left(\sigma^2\right)$, with probability at least $1 - \delta'$ we have $\|X\|_2 \leq 4\sigma\sqrt{d \log \frac{1}{\delta'}}$.*

**Lemma 10** ([47]). *Let $X_1, X_2, \cdots, X_n$ be $n$ (zero mean) random variables such that each $X_i$ is sub-Gaussian with $\sigma^2$. Then the following holds*

$$\mathbb{P}\left(\max_{i \in n} X_i \geq t\right) \leq n e^{-\frac{t^2}{2\sigma^2}},$$

$$\mathbb{P}\left(\max_{i \in n} |X_i| \geq t\right) \leq 2n e^{-\frac{t^2}{2\sigma^2}}.$$

Below is a lemma related to the Gaussian random variable. We will employ it to bound the noise added by the Gaussian mechanism.

**Lemma 11.** *Let $\{x_1, \cdots, x_n\}$ be $n$ random variables sampled from Gaussian distribution $\mathcal{N}\left(0, \sigma^2\right)$. Then*

$$\mathbb{E}\left[\max_{1 \leq i \leq n} |x_i|\right] \leq \sigma\sqrt{2\log 2n},$$

$$\mathbb{P}\left(\left\{\max_{1 \leq i \leq n} |x_i| \geq t\right\}\right) \leq 2n e^{-\frac{t^2}{2\sigma^2}}.$$

*Particularly, if $n = 1$, we have $\mathbb{P}\left(\{|x_i| \geq t\}\right) \leq 2 e^{-\frac{t^2}{2\sigma^2}}$.*

**Lemma 12** (Hoeffding's inequality). *Let $X_1, \cdots, X_n$ be independent random variables bounded by the interval $[a, b]$. Then, for any $t > 0$,*

$$\mathbb{P}\left(\left|\frac{1}{n}\sum_{i=1}^{n} X_i - \frac{1}{n}\sum_{i=1}^{n} \mathbb{E}[X_i]\right| > t\right) \leq 2\exp\left(-\frac{2nt^2}{(b-a)^2}\right).$$

**Lemma 13** (Bernstein's inequality for bounded random variables). *Let $X_1, X_2, \cdots, X_n$ be independent centered bounded random variables, i.e. $|X_i| \leq M$ and $\mathbb{E}[X_i] = 0$, with variance $\mathbb{E}[X_i^2] = \sigma^2$. Then, for any $t > 0$,*

$$\mathbb{P}\left(\left|\sum_{i=1}^{n} X_i\right| > \sqrt{2n\sigma^2 t} + \frac{2Mt}{3}\right) \leq 2e^{-t}.$$

**Lemma 14** (Bound on threshold $\tau_{\alpha,\beta}$). *Under the Assumption 5, $\tau_{\alpha,\beta} \leq \frac{1}{\lambda}\log\frac{1}{\alpha\beta}$.*

**Lemma 15.** *For any $w, w' \in \mathbb{R}^d$ and closed convex set $\mathcal{C} \subseteq \mathbb{R}^d$ we have*

$$\|\Pi_{\mathcal{C}}(w) - \Pi_{\mathcal{C}}(w')\|_2 \leq \|w - w'\|_2,$$

*where $\Pi_{\mathcal{C}}$ is the projection operation onto the set $\mathcal{C}$, i.e., $\Pi_{\mathcal{C}}(v) = \arg\min_{u \in \mathcal{C}} \|u - v\|_2$.*

## E.1 Lemmas for privacy guarantee

**Lemma 16** (Post-processing). *Let $\mathcal{M} : \mathcal{X}^n \to \mathcal{Y}$ be an $\varepsilon$-differential private mechanism. Consider $F : \mathcal{Y} \to \mathcal{Z}$ as an arbitrary randomized mapping. Then the mechanism $F \circ \mathcal{M}$ is $\varepsilon$-differential private.*

**Lemma 17** (Billboard lemma [26]). *Let $\mathcal{M} : \mathcal{D}^n \to \mathcal{O}$ be an $(\varepsilon, \delta)$-differential private mechanism. Consider a set of $n$ functions $\pi_i : \mathcal{D} \times \mathcal{O} \to \mathcal{R}$, for $i \in [n]$. Then the mechanism $\mathcal{M}' : \mathcal{D}^n \to \mathcal{O} \times \mathcal{R}^n$ that computes $r = \mathcal{M}(D)$ and outputs $\mathcal{M}'(D) = (r, \pi_1(D_1, r), \cdots, \pi_n(D_n, r))$, where $D_i$ is the agent $i$'s data, is $(\varepsilon, \delta)$-joint differential private.*

**Theorem 18** (Parallel Composition Theorem). *Let $O_1, O_2, ..., O_k$ be $n$ independent operations that satisfy $(\varepsilon_i, \delta$-DP for each $i \in [k]$, then the parallel composition of these operations guarantee $(\varepsilon_1 + \varepsilon_2 + ... + \varepsilon_k, \delta)$-DP.*

**Theorem 19** (Sequential Composition Theorem(Theorem 4 in [33])). *If $O_1, O_2, ..., O_k$ are sequential operations that satisfy $\varepsilon$-individual differential privacy with a failure probability of $\delta$, then the sequential composition of these operations guarantees $(\varepsilon_1 + \varepsilon_2 + ... + \varepsilon_k, k \cdot \delta)$-individual differential privacy, where $\varepsilon_i$ represents the $\varepsilon$ of operation $O_i$, and $k$ is the total number of operations performed.*

## E.2 Technical lemmas for upper bounding the accuracy

**Lemma 20.** *Let $\hat{\theta}(D)$ and $\hat{\theta}(D')$ be the estimators on two fixed datasets $D, D'$ that differ on at most $k$ entries and let $\hat{D}$ denote the fixed dataset that differs from $D$ on at most one entries. Suppose that with probability at least $1 - \gamma_n$, if $\|\hat{\theta}(D) - \hat{\theta}(\hat{D})\|_2 \le \lambda_n$ then we have with probability at least $1 - g\gamma_n$, it holds that*

$$\|\hat{\theta}(D) - \hat{\theta}(D')\|_2 \le g\lambda_n.$$

**Lemma 21** (Theorem 2 in [59]). *Suppose we solve the problem of the form $\min_\theta \|\theta - \bar{\theta}\|_2^2 + \lambda_n \|\theta\|_1$, such that constraint term $\lambda_n$ is set as $\lambda_n \ge \left\| \theta^* - \bar{\theta}(D) \right\|_\infty$. Then, the optimal solution $\hat{\theta} = S_{\lambda_n}(\bar{\theta})$ satisfies:*

$$\left\| \hat{\theta}(D) - \theta^* \right\|_\infty \le 2\lambda_n,$$

$$\left\| \hat{\theta}(D) - \theta^* \right\|_2 \le 4\sqrt{k}\lambda_n,$$

$$\left\| \hat{\theta}(D) - \theta^* \right\|_1 \le 8k\lambda_n.$$

**Lemma 22.** *Under Assumption 1 and 6, we set $\tau_x = \Theta(\frac{\sigma\sqrt{\log n}}{\sqrt{d}}), \tau_y = \Theta(\sigma\sqrt{\log n}), r = \Theta(\sigma\sqrt{\log n}), \lambda_n = O\left( \frac{r^2 \sqrt{\log d \log \frac{1}{\delta}}}{\sqrt{n}\varepsilon} \right)$, when $n$ is sufficient large such that $n \ge \Omega(\frac{s^2 r^4 \log d \log \frac{1}{\delta}}{\varepsilon^2 \kappa_\infty})$, with probability at least $1 - O(d^{-\Omega(1)})$, one has*

$$\left\| \hat{\theta}^P(D) - \theta^* \right\|_2 \le \sqrt{k}\lambda_n$$

**Lemma 23.** *(Weyl's Inequality[42]) Let $X, Y \in \mathbb{R}^{d \times d}$ be two symmetric matrices, and $E = X - Y$. Then, for all $i = 1, \cdots, d$, we have*

$$|\lambda_i(X) - \lambda_i(Y)| \le \|E\|_2,$$

*where we take some liberties with the notation and use $\lambda_i(M)$ to denote the $i$-th eigenvalue of the matrix $M$.*

## E.3 Technical lemmas for covariance matrix estimation

**Lemma 24** ([4]). *If $\{x_1, x_2, \cdots, x_n\}$ are $n$ realizations of a (zero mean) $\sigma^2$-sub-Gaussian random vector $X$ with covariance matrix $\Sigma_{XX} = \mathbb{E}[XX^T]$, and $\hat{\Sigma}_{\bar{X}\bar{X}} = \left( \hat{\sigma}_{\bar{x}\bar{x}^T, ij} \right)_{1 \le i, j \le d} = \frac{1}{n} \sum_{i=1}^n \bar{x}_i \bar{x}_i^T$ is the empirical covariance matrix, then there exist constants $C_1$ and $\gamma > 0$ such that for any $i, j \in [d]$, we have:*

$$\mathbb{P}\left( \left\| \hat{\Sigma}_{\bar{X}\bar{X}} - \Sigma_{XX} \right\|_{\infty, \infty} > t \right) \le C_1 e^{-nt^2 \frac{8}{\gamma^2}},$$

*for all $|t| \le \phi$ with some $\phi$, where $C_1$ and $\gamma$ are constants and depend only on $\sigma^2$. Specifically,*

$$\mathbb{P}\left( \left\| \hat{\Sigma}_{\bar{X}\bar{X}} - \Sigma_{XX} \right\|_{\infty,\infty} \ge \gamma \sqrt{\frac{\log d}{n}} \right) \le C_1 d^{-8}.$$

**Lemma 25.** *For every fixed $1 \le i, j \le d$, there exists a constant $C_1 > 0$ such that with probability at least $1 - C_1 d^{-\frac{9}{2}}$, the following holds:*

$$
\begin{aligned}
&\left| \ddot{\sigma}_{\bar{x}\bar{x}^T, ij} - \sigma_{xx^T, ij} \right| \\
&\le 4 \min \left\{ \left| \sigma_{xx^T, ij} \right|, \gamma \sqrt{\frac{\log d}{n}} + \frac{4r^2 \sqrt{2 \ln 1.25/\delta} \sqrt{\log d}}{n\varepsilon} \right\}
\end{aligned}
\tag{5}
$$

# F   Proofs of Supporting Lemmas

**Proof of Lemma 15.** Denote $b = \Pi_{\mathcal{C}}(w)$ and $b' = \Pi_{\mathcal{C}}(w')$. Since $b$ and $b'$ are in $\mathcal{C}$, so the segment $bb'$ is contained in $\mathcal{C}$, thus we have for all $t \in [0,1]$, $\|(1-t)b + tb' - w\|_2 \ge \|b - w\|_2$. Thus

$$0 \le \frac{d}{dt} \|tb + (1-t)b' - w\|_2^2|_{t=0} = 2\langle b' - b, b - w \rangle$$

Similarly, we have $\langle b - b', b' - w' \rangle \ge 0$. Now consider the function $D(t) = \|(1-t)b + tw - (1-t)b' - tw'\|_2^2 = \|b - b' + t(w - w' + b' - b)\|_2^2$, which is a quadratic function in $t$. And by the previous two inequalities we have $D'(0) = 2\langle b - b', w - w' + b' - b \rangle \ge 0$. Thus $D(\cdot)$ is a increasing function on $[0, 2)$, thus $D(1) \ge D(0)$ which means $\|w - w'\|_2 \ge \|b - b'\|_2$. $\qquad\square$

**Proof of Lemma 14.** We first bound $\tau_{\alpha,\beta}^1$. Since $n = \#\{i : c_i \le \tau\} + \#\{i : c_i > \tau\}$, the event $\{\#\{i : c_i \le \tau\} \ge (1-\alpha)n\}$ is equivalent to the event $\{\#\{i : c_i > \tau\} \le \alpha n\}$. Thus, by the definition of $\tau_{\alpha,\beta}^1$,

$$
\begin{aligned}
\tau_{\alpha,\beta}^1 &= \inf\{\tau > 0 : \mathbb{P}_{(c_1,\cdots,c_n)\sim p^n}(\#\{i : c_i > \tau\} \le \alpha n) \ge 1 - \beta\} \\
&= \inf\{\tau > 0 : \mathbb{P}_{(c_1,\cdots,c_n)\sim p^n}(\#\{i : c_i > \tau\} > \alpha n) \le \beta\},
\end{aligned}
$$

By Markov's inequality, we have

$$
\begin{aligned}
\mathbb{P}_{(c_1,\cdots,c_n)\sim p^n}(\#\{i : c_i > \tau\} > \alpha n) &\le \frac{\mathbb{E}_{(c_1,\cdots,c_n)\sim p^n}[\sum_{i=1}^n I_{\{c_i > \tau\}}]}{\alpha n} \\
&= \frac{\sum_{i=1}^n \mathbb{E}_{c_i \sim p}[I_{\{c_i > \tau\}}]}{\alpha n} = \frac{n\mathbb{P}[c_i > \tau]}{\alpha n} = \frac{\mathbb{P}[c_i > \tau]}{\alpha}.
\end{aligned}
$$

Thus, $\{\tau > 0 : \mathbb{P}(c_i > \tau) \le \alpha\beta\} \subseteq \{\tau > 0 : \mathbb{P}_{(c_1,\cdots,c_n)\sim p^n}(\#\{i : c_i > \tau\} > \alpha n) \le \beta\}$, which implies $\tau_{\alpha,\beta}^1 \le \inf\{\tau > 0 : \mathbb{P}(c_i > \tau) \le \alpha\beta\}$. The Assumption 5 implies that $\mathbb{P}(c_i > \tau) \le e^{-\lambda\tau}$. Hence, $\tau_{\alpha,\beta}^1 \le \frac{1}{\lambda} \log \frac{1}{\alpha\beta}$. By the definition of $\tau_\alpha^2$ and Assumption 5, we have $\tau_\alpha^2 \le \frac{1}{\lambda} \ln \frac{1}{\alpha}$. Since $\beta \in (0,1)$, $\frac{1}{\lambda} \log \frac{1}{\alpha\beta} > \frac{1}{\lambda} \log \frac{1}{\alpha}$, then $\tau_{\alpha,\beta} = \max\{\tau_{\alpha,\beta}^1, \tau_\alpha^2\} \le \frac{1}{\lambda} \log \frac{1}{\alpha\beta}$. $\qquad\square$

**Proof of Lemma 4.** For the ease of presentation, we denote the initial parameter $(\ddot{\Sigma}_{\bar{X}\bar{X}}^{-1})\dot{\Sigma}_{\widetilde{X}\widetilde{Y}}$ on which the soft-thresholding $S_{\lambda_n}$ is performed by $\bar{\theta}$. Let $\Delta$ be the error vector $\Delta = \hat{\theta}^P(\hat{D}) - \hat{\theta}^P(\hat{D})$. It follows that

$$
\begin{aligned}
\|\Delta\|_\infty &= \|\hat{\theta}^P(\hat{D}) - \bar{\theta}(\hat{D}) + \bar{\theta}(\hat{D}) - \bar{\theta}(D) + \bar{\theta}(D) - \hat{\theta}^P(D)\|_\infty \\
&\le \|\hat{\theta}^P(\hat{D}) - \bar{\theta}(\hat{D})\|_\infty + \|\bar{\theta}(\hat{D}) - \bar{\theta}(D)\|_\infty + \|\bar{\theta}(D) - \hat{\theta}^P(D)\|_\infty
\end{aligned}
\tag{6}
$$

where we utilize the fact that $\hat{\theta}^P(\hat{D})$ is feasible.

Using the property of generalized thresholding operator, we can bound the first term of equation 6 as $\|\hat{\theta}^P(\hat{D}) - \bar{\theta}(\hat{D})\|_\infty \le \lambda_n$. For the third term of equation 6, from Lemma 22, we know that $\|\bar{\theta}(D) - \hat{\theta}^P(D)\|_\infty \le \lambda_n$.

Our next task it to show $\lambda_n$ is indeed greater than the second term of equation 6 $\|\bar{\theta}(\hat{D}) - \bar{\theta}(D)\|_\infty$. It can be expressed as

$$\|\bar{\theta}(\hat{D}) - \bar{\theta}(D)\|_\infty = \| \left( \ddot{\Sigma}_{\bar{X}\bar{X}}(\hat{D}) \right)^{-1} \dot{\Sigma}_{\widetilde{X}\widetilde{Y}}(\hat{D}) - \left( \ddot{\Sigma}_{\bar{X}\bar{X}}(D) \right)^{-1} \dot{\Sigma}_{\widetilde{X}\widetilde{Y}}(D) \|_\infty.$$

Applying the inequality $\|AB - A'B'\|_\infty = \|AB - AB' + AB' - A'B'\|_\infty \le \|A\|_\infty \|B - B'\|_\infty + \|A - A'\|_\infty \|B'\|_\infty$, we have,

$$\begin{aligned}
& \| \left( \ddot{\Sigma}_{\bar{X}\bar{X}}(\hat{D}) \right)^{-1} \dot{\Sigma}_{\widetilde{X}\widetilde{Y}}(\hat{D}) - \left( \ddot{\Sigma}_{\bar{X}\bar{X}}(D) \right)^{-1} \dot{\Sigma}_{\widetilde{X}\widetilde{Y}}(D) \|_\infty \\
\le & \| \left( \ddot{\Sigma}_{\bar{X}\bar{X}}(\hat{D}) \right)^{-1} \|_\infty \| \dot{\Sigma}_{\widetilde{X}\widetilde{Y}}(\hat{D}) - \dot{\Sigma}_{\widetilde{X}\widetilde{Y}}(D) \|_\infty \\
& + \| \left( \ddot{\Sigma}_{\bar{X}\bar{X}}(\hat{D}) \right)^{-1} - \left( \ddot{\Sigma}_{\bar{X}\bar{X}}(D) \right)^{-1} \|_\infty \| \dot{\Sigma}_{\widetilde{X}\widetilde{Y}}(D) \|_\infty
\end{aligned}$$

(7)

For the first term of equation 7, we see from Lemma 1 that when $n \ge \Omega \left( \frac{s^2 r^4 \log d \log \frac{1}{\delta}}{\varepsilon^2 \kappa_\infty} \right)$, $\left\| \left( \ddot{\Sigma}_{\bar{X}\bar{X}} \right)^{-1} \right\|_\infty \le \frac{2}{\kappa_\infty}$. And by equation 15, we have with probability $1 - 2d^{-8}$ that

$$\begin{aligned}
& \| \dot{\Sigma}_{\widetilde{X}\widetilde{Y}}(\hat{D}) - \dot{\Sigma}_{\widetilde{X}\widetilde{Y}}(D) \|_\infty \\
= & \| \hat{\Sigma}_{\widetilde{X}\widetilde{Y}}(\hat{D}) - N_2(\hat{D}) + N_2(D) - \hat{\Sigma}_{\widetilde{X}\widetilde{Y}}(D) \|_\infty \\
\le & \| \hat{\Sigma}_{\widetilde{X}\widetilde{Y}}(\hat{D}) - \hat{\Sigma}_{\widetilde{X}\widetilde{Y}}(D) \|_\infty + \| N_2(\hat{D}) \|_\infty + \| N_2(D) \|_\infty \\
\le & \frac{1}{n} \max_i (\|x_i\|_2(|\tilde{y}_i|) + \|x_i\|_2(|\tilde{y}_i|)) + \frac{8 r \tau_y \sqrt{2 \log \frac{1}{\delta} \log d}}{\varepsilon n} \\
\le & \frac{2 r \tau_y}{n} + \frac{8 r \tau_y \sqrt{2 \log \frac{1}{\delta} \log d}}{\varepsilon n} \\
\le & O \left( \frac{r \tau_y \sqrt{\log \frac{1}{\delta} \log d}}{\varepsilon n} \right) \le \lambda_n
\end{aligned}$$

For the second term of equation 7, note that for any two nonsingular square matrices $A, B$ with the same size, it holds that $A^{-1} - B^{-1} = -B^{-1}(A - B)A^{-1}$. Thus, by Lemma 1, we have

$$\begin{aligned}
& \| \left( \ddot{\Sigma}_{\bar{X}\bar{X}}(\hat{D}) \right)^{-1} - \left( \ddot{\Sigma}_{\bar{X}\bar{X}}(D) \right)^{-1} \|_\infty \\
\le & \| \left( \ddot{\Sigma}_{\bar{X}\bar{X}}(\hat{D}) \right)^{-1} \|_\infty \| - \left( \ddot{\Sigma}_{\bar{X}\bar{X}}(D) \right)^{-1} \|_\infty \| \left( \ddot{\Sigma}_{\bar{X}\bar{X}}(\hat{D}) \right) - \left( \ddot{\Sigma}_{\bar{X}\bar{X}}(D) \right) \|_\infty \\
\le & \frac{4}{\kappa_\infty^2} \left( \| \ddot{\Sigma}_{\bar{X}\bar{X}}(\hat{D}) - \Sigma \|_\infty + \| \ddot{\Sigma}_{\bar{X}\bar{X}}(D) - \Sigma \|_\infty \right) \\
\le & c_2 \left( \frac{s r^2 \sqrt{\log d \log(\frac{1}{\delta})}}{\kappa_\infty^2 \varepsilon \sqrt{n}} \right) \le \lambda_n
\end{aligned}$$

Combining above, we have

$$\begin{aligned}
& \| \hat{\theta}^P(\hat{D}) - \hat{\theta}^P(D) \|_\infty \\
\le & 2\lambda_n + O \left( \frac{r \tau_y \sqrt{\log \frac{1}{\delta} \log d}}{\varepsilon n} \right) + O \left( \frac{s r^2 \sqrt{\log d \log(\frac{1}{\delta})}}{\kappa_\infty^2 \varepsilon \sqrt{n}} \right) \\
\le & 4\lambda_n
\end{aligned}$$

(8)

To further our analysis, we introduce the notion of decomposibility of norm and subspace compatibility constant. We call $\left(\mathcal{M}, \overline{\mathcal{M}}^{\perp}\right)$ a subspace pairs where $\mathcal{M}$ is the model subspace in which the estimated model parameter $\theta^*$ and similarly structured parameters lie, and which is typically low-dimensional, while $\overline{\mathcal{M}}^{\perp}$ is the perturbation subspace of parameters that represents perturbations away from the model subspace.

**Definition 12.** *A regularization function $\mathcal{R}$ is said to be decomposable with respect to a subspace pair $\left(\mathcal{M}, \overline{\mathcal{M}}^{\perp}\right)$, if $\mathcal{R}(u + v) = \mathcal{R}(u) + \mathcal{R}(v)$, for all $u \in \mathcal{M}, v \in \overline{\mathcal{M}}^{\perp}$.*

Note that when $\mathcal{R}(\cdot)$ is a norm, by the triangle inequality, the LHS is always less than or equal to the RHS, so that the equality indicates the largest possible value for the LHS. For notational simplicity, we use $(S, S^c)$ instead of an arbitrary subspace pair $\left(\mathcal{M}, \overline{\mathcal{M}}^{\perp}\right)$.

**Definition 13.** *The subspace compatibility constant is defined as $\mathcal{M}$ : $\Psi(\mathcal{M}, \|\cdot\|)$ := $\sup_{u \in \mathcal{M} \setminus \{0\}} \frac{\mathcal{R}(u)}{\|u\|}$.*

It is noted that subspace compatibility constant that captures the relationship between the regularization function $\mathcal{R}(\cdot)$ and the error norm $\|\cdot\|$, over vectors in subspace. In our proof, it is clear that the regularization is the $\ell_1$-norm, which is decomposable with respect to the subspace pair. We also denote the subspace of vectors in $\mathbb{R}^d$ by $S$. Since we assume $k$ is the sparsity level of $\theta^*$, the cardinality of the support set of the model space where the true parameter $\theta^*$ lies. It can be seen that any parameter $\theta \in S$ would be at-most $k$-sparse. Therefore, we have that $\Psi\left(S, \|\cdot\|_2\right) \leq \sqrt{k}$.

Additionally, we use the notion $\Pi_S(\Delta)$ to represent the $\ell_2$ projection onto the model space $S$. Then, by the assumption of the statement that $\theta^*_{S^c} = 0$, and the decomposability of $\|\cdot\|_1$ with respect to $(S, S^c)$,

$$
\begin{aligned}
\|\hat{\theta}^P(\hat{D})\|_1 &= \|\hat{\theta}^P(\hat{D})\|_1 + \|\Pi_{S^c}(\Delta)\|_1 - \|\Pi_{S^c}(\Delta)\|_1 \\
&= \|\hat{\theta}^P(\hat{D}) + \Pi_{S^c}(\Delta)\|_1 - \|\Pi_{S^c}(\Delta)\|_1 \\
&\overset{(i)}{\leq} \|\hat{\theta}^P(\hat{D}) + \Pi_{S^c}(\Delta) + \Pi_S(\Delta)\|_1 + \|\Pi_S(\Delta)\|_1 - \|\Pi_{S^c}(\Delta)\|_1 \\
&= \|\hat{\theta}^P(\hat{D}) + \Delta\|_1 + \|\Pi_S(\Delta)\|_1 - \|\Pi_{S^c}(\Delta)\|_1
\end{aligned} \tag{9}
$$

where the equality $(i)$ holds by the triangle inequality, which is the basic property of norms. Since we are minimizing the objective function $\|\hat{\theta}^P(D)\|_1$, we obtain the inequality of $\|\hat{\theta}^P(\hat{D}) + \Delta\|_1 = \|\hat{\theta}^P(D)\|_1 \leq \|\hat{\theta}^P(\hat{D})\|_1$. Combining this inequality with equation 9, we have

$$
0 \leq \|\Pi_S(\Delta)\|_1 - \|\Pi_{S^c}(\Delta)\|_1 \tag{10}
$$

Armed with inequalities equation 8 and equation 10, we utilize the Hölder's inequality and the decomposability of our regularizer $\|\cdot\|_1$ in order to derive the error bounds in terms of $\ell_2$ norm:

$$
\begin{aligned}
\|\Delta\|_2^2 = \langle \Delta, \Delta \rangle &\leq \|\Delta\|_\infty \|\Delta\|_1 \\
&\leq \|\Delta\|_\infty \left( \|\Pi_S(\Delta)\|_1 + \|\Pi_{S^c}(\Delta)\| \right) \|_1.
\end{aligned}
$$

Since the error vector $\Delta$ satisfies the inequality equation 10,

$$
\|\Delta\|_2^2 \leq 2\|\Delta\|_\infty \|\Pi_S(\Delta)\|_1 \tag{11}
$$

Combining all the pieces together yields

$$
\|\Delta\|_2^2 \leq 4\Psi(S)\lambda_n \|\Pi_S(\Delta)\|_2
$$

where $\Psi(\mathcal{M})$ is the abbreviation for $\Psi\left(S, \|\cdot\|_2\right)$.

Notice that the projection operator is non-expansive, $\|\Pi_{\mathcal{S}}(\Delta)\|_2^2 \leq \|\Delta\|_2^2$. Hence, we obtain $\|\Pi_{\mathcal{S}}(\Delta)\|_2 \leq 4\Psi(S)\lambda_n$, and plugging it back into equation 11 yields the $\ell_2$ error bounds.

$\square$

**Proof of Lemma 20.** Define a sequence of datasets $D^0, D^1, \cdots, D^k$, such that $D^0 = D$, $D^k = D'$, and for each $i \in [k]$, $D^i, D^{i-1}$ differ on at most one agent's dataset. Then, by the triangular inequality, we obtain

$$\|\hat{\theta}(D) - \hat{\theta}(D')\|_2 = \|\hat{\theta}(D^0) - \hat{\theta}(D^k)\|_2 = \|\sum_{i=1}^{k} \hat{\theta}(D^{i-1}) - \hat{\theta}(D^i)\|_2 \leq \sum_{i=1}^{k} \|\hat{\theta}(D^{i-1}) - \hat{\theta}(D^i)\|_2 \leq g\Delta_n.$$

with probability at least $1 - k\gamma_n$ by taking a union bound over $g$ failure probabilities $\gamma_n$. $\square$

**Proof of Lemma 22.** Before giving theoretical analysis, we first prove that $\ddot{\Sigma}_{\bar{X}\bar{X}}$ is invertible with high probability. With the help of Weyl's Inequality (Lemma 23), we can see that to show $\ddot{\Sigma}_{\bar{X}\bar{X}}$ is invertible it is sufficient to show that $\|\ddot{\Sigma}_{\bar{X}\bar{X}} - \Sigma\|_2 \leq \frac{\lambda_{\min}(\Sigma)}{2}$. This is due to that by Lemma 23, we have

$$\lambda_{\min}(\Sigma) - \|\ddot{\Sigma}_{\bar{X}\bar{X}} - \Sigma\|_2 \leq \lambda_{\min}(\ddot{\Sigma}_{\bar{X}\bar{X}}).$$

Thus, if $\|\ddot{\Sigma}_{\bar{X}\bar{X}} - \Sigma\|_2 \leq \frac{\lambda_{\min}(\Sigma)}{2}$, we have $\lambda_{\min}(\ddot{\Sigma}_{\bar{X}\bar{X}}) \geq \frac{\lambda_{\min}(\Sigma)}{2} > 0$.

Thus, by Lemma 1, it is sufficient to show that $\lambda_{\min}(\Sigma) \geq O\left(\frac{sr^2\sqrt{\log d \log \frac{1}{\delta}}}{\varepsilon\sqrt{n}}\right)$, which is true under the assumption of $n \geq \Omega\left(\frac{s^2 r^4 \log d \log \frac{1}{\delta}}{\varepsilon^2 \lambda_{\min}(\Sigma)^2}\right)$. Thus, with probability at least $1 - \exp(-\Omega(d)) - \xi$, it is invertible. In the following we will always assume that this event holds.

To prove the theorem, we first introduce the following lemma on the estimation error of $\hat{\theta}$ in equation 2.

Note that this is a non-probabilistic result, and it holds deterministically for any selection of $\lambda_n$ or any distributional setting of the covariates $x_i$. Our goal is to show that $\lambda_n \geq \left\|\theta^* - \left(\ddot{\Sigma}_{\bar{X}\bar{X}}\right)^{-1}\left(\ddot{\Sigma}_{\tilde{X}\tilde{Y}}\right)\right\|_\infty$ under the assumptions specified in Lemma 21.

$$\begin{aligned}
\left\|\theta^* - \hat{\theta}^P(D)\right\|_\infty &= \left\|\theta^* - \left(\ddot{\Sigma}_{\bar{X}\bar{X}}\right)^{-1}\left(\ddot{\Sigma}_{\tilde{X}\tilde{Y}}\right)\right\|_\infty \\
&\leq \left\|\left(\ddot{\Sigma}_{\bar{X}\bar{X}}\right)^{-1}\right\|_\infty \left\|\left(\ddot{\Sigma}_{\bar{X}\bar{X}}\right)\theta^* - \left(\widehat{\Sigma}_{\tilde{X}\tilde{Y}} + N_2\right)\right\|_\infty
\end{aligned} \tag{12}$$

where the vector $N_2 \in \mathbb{R}^d$ is sampled from $\mathcal{N}(0, \frac{32\tau_x^2\tau_y^2 \log \frac{1.25}{\delta}}{n^2\varepsilon^2} I_d)$. We first develop upper bound of $\ddot{\Sigma}_{\bar{X}\bar{X}}$. For any nonzero vector $w \in \mathbb{R}^d$, Note that

$$\begin{aligned}
\left\|\ddot{\Sigma}_{\bar{X}\bar{X}} w\right\|_\infty &= \left\|\ddot{\Sigma}_{\bar{X}\bar{X}} w - \Sigma w + \Sigma w\right\|_\infty \\
&\geq \|\Sigma w\|_\infty - \left\|\left(\ddot{\Sigma}_{\bar{X}\bar{X}} - \Sigma\right) w\right\|_\infty \\
&\geq \left(\kappa_\infty - \left\|\ddot{\Sigma}_{\bar{X}\bar{X}} - \Sigma\right\|_\infty\right) \|w\|_\infty.
\end{aligned}$$

Our objective is to find a sufficiently large $n$ such that $\left\|\ddot{\Sigma}_{\bar{X}\bar{X}} - \Sigma\right\|_\infty$ is less than $\frac{\kappa_\infty}{2}$.

By Lemma 1 we can see the following:

$$\begin{aligned}
\|\ddot{\Sigma}_{\bar{X}\bar{X}} - \Sigma\|_\infty^2 &= \|\ddot{\Sigma}_{\bar{X}\bar{X}} - \Sigma\|_1^2 \\
&\leq O\left(s^2 \left(\frac{\log d \log \frac{1}{\delta} r^4}{n\varepsilon^2}\right)^{1-q} + \left(\frac{\log d \log \frac{1}{\delta} r^4}{n\varepsilon^2}\right)\right)
\end{aligned} \tag{13}$$

Thus, when $n \geq \Omega\left(\frac{s^2 r^4 \log d \log \frac{1}{\delta}}{\varepsilon^2 \kappa_\infty}\right)$, we have $\left\|\ddot{\Sigma}_{\bar{X}\bar{X}} w\right\|_\infty \geq \frac{\kappa_\infty}{2}\|w\|_\infty$, which implies $\left\|\left(\ddot{\Sigma}_{\bar{X}\bar{X}}\right)^{-1}\right\|_\infty \leq \frac{2}{\kappa_\infty}$.

Given sufficiently large $n$, from equation 12, we have:

$$\left\|\theta^* - \hat{\theta}^P(D)\right\|_\infty$$

$$\leq \frac{2}{\kappa_\infty} \left\|\left(\ddot{\Sigma}_{\bar{X}\bar{X}}\right)\theta^* - \left(\widehat{\Sigma}_{\widetilde{X}\widetilde{Y}} + N_2\right)\right\|_\infty \tag{14}$$

$$\leq \frac{2}{\kappa_\infty}\left\{ \underbrace{\left\|\widehat{\Sigma}_{\widetilde{X}\widetilde{Y}} - \Sigma_{\widetilde{X}\widetilde{Y}}\right\|_\infty}_{T_1} + \underbrace{\left\|\Sigma_{\widetilde{X}\widetilde{Y}} - \Sigma_{YX}\right\|_\infty}_{T_2} + \underbrace{\left\|\left(\ddot{\Sigma}_{\bar{X}\bar{X}} - \Sigma\right)\theta^*\right\|_\infty}_{T_3} + \underbrace{\|N_2\|_\infty}_{N_2} \right\}$$

We will bound the above four terms one by one.

For $1 \leq j \leq d$, we have $\operatorname{Var}(\tilde{y}_i \tilde{x}_{ij}) \leq \operatorname{E}(\tilde{y}_i \tilde{x}_{ij})^2 \leq \operatorname{E}(y_i x_{ij})^2 \leq \sqrt{\operatorname{E}y_i^4 \operatorname{E}x_{ij}^4} =: v_1 < \infty$. In addition, $\operatorname{E}|\tilde{y}_i \tilde{x}_{ij}|^p \leq (\tau_x \tau_y)^{p-2} v_1$ holds. Therefore, according to Lemma 13, we have:

$$P\left(\left|\widehat{\sigma}_{\widetilde{Y}\widetilde{x}_j} - \sigma_{\widetilde{Y}\widetilde{x}_j}\right| \geq \sqrt{\frac{2v_1 t}{n}} + \frac{c\tau_x \tau_y t}{n}\right) \leq \exp(-t),$$

where $\widehat{\sigma}_{\widetilde{Y}\widetilde{x}_j} = \frac{1}{n}\sum_{i=1}^n \tilde{y}_i \tilde{x}_{ij}, \sigma_{\widetilde{Y}\widetilde{x}_j} = \operatorname{E}\tilde{y}_i \tilde{x}_{ij}$ and $c$ is a certain constant. Then by the union bound, the following can be derived:

$$P\left(|T_1| > \sqrt{\frac{2v_1 t}{n}} + \frac{c\tau_x \tau_y t}{n}\right) \leq d\exp(-t).$$

Next, we give an estimation of $T_2$. Note that for $1 \leq j \leq d$, by lemma 9 we have:

$$\operatorname{E}\tilde{y}_i \tilde{x}_{ij} - \operatorname{E}y_i x_{ij} = \operatorname{E}\tilde{y}_i \tilde{x}_{ij} - \operatorname{E}\tilde{y}_i x_{ij} + \operatorname{E}\tilde{y}_i x_{ij} - \operatorname{E}y_i x_{ij}$$

$$= \operatorname{E}\tilde{y}_i\left(\tilde{x}_{ij} - x_{ij}\right) + \operatorname{E}\left(\tilde{y}_i - y_i\right)x_{ij}$$

$$\leq \sqrt{\operatorname{E}\left(y_i^2\left(\tilde{x}_{ij} - x_{ij}\right)^2\right)P\left(|x_{ij}| \geq \tau_x\right)} + \sqrt{\operatorname{E}\left(\left(\tilde{y}_i - y_i\right)^2 x_{ij}^2\right)P\left(|y_i| \geq \tau_y\right)}$$

$$\leq \sqrt{v_1}\left(2e^{-\frac{\tau_x^2}{2\sigma^2}} + 2e^{-\frac{\tau_y^2}{2\sigma^2}}\right),$$

which shows that $T_2 \leq \sqrt{v_1}\left(2e^{-\frac{\tau_x^2}{2\sigma^2}} + 2e^{-\frac{\tau_y^2}{2\sigma^2}}\right)$.

To upper bound term $T_3$, we need to evaluate $\|\ddot{\Sigma}_{\bar{X}\bar{X}} - \Sigma\|_\infty$ and we can reuse obtained results from Lemma 1.

We can see that $T_3$ is bounded by $O(\frac{\sqrt{\log d \log \frac{1}{\delta}}}{\sqrt{n}\varepsilon})$. Here we used the fact that $\ddot{\Sigma}_{\bar{X}\bar{X}} - \Sigma$ is a symmetric matrix.

$$\|(\ddot{\Sigma}_{\bar{X}\bar{X}} - \Sigma)\theta^*\|_\infty \leq \|\ddot{\Sigma}_{\bar{X}\bar{X}} - \Sigma\|_2\|\theta^*\|_2 \leq \|\ddot{\Sigma}_{\bar{X}\bar{X}} - \Sigma\|_1\|\theta^*\|_2$$

$$\leq O(r^2\sqrt{\left(\frac{\log d \log \frac{1}{\delta}}{n\varepsilon^2}\right)})\|\theta^*\|_2,$$

given the selection of $r$.

The last term of equation 14 can be bounded by Gaussian tail bound by lemma 11. With probability $1 - O(d^{-8})$, we have:

$$\|N_2\|_\infty \leq O\left(\frac{r\tau_y\sqrt{\log\frac{1}{\delta}\log d}}{\varepsilon n}\right). \tag{15}$$

Finally combining all pieces, we can find that $T_3$ is the dominating term. Since $\lambda_n \geq \left\| \theta^* - \left( \ddot{\Sigma}_{\bar{X}\bar{X}} \right)^{-1} \left( \ddot{\Sigma}_{\widetilde{X}\widetilde{Y}} \right) \right\|_\infty$, Lemma 21 implies that with probability at least $1 - O(d^{-8}) - e^{-\Omega(d)}$,

$$\left\| \theta^* - \left[ \ddot{\Sigma}_{\bar{X}\bar{X}} \right]^{-1} \left( \widehat{\Sigma}_{\widetilde{X}\widetilde{Y}} + N_2 \right) \right\|_2 \leq O \left( \frac{\sqrt{k \log d \log \frac{1}{\delta}}}{\sqrt{n}\varepsilon} \right),$$

which completes our proof of Theorem. $\qquad\qquad\square$

**Proof of Lemma 1.** Our goal is to prove

$$\mathbb{E}\|\ddot{\Sigma}_{\bar{X}\bar{X}} - \Sigma\|^2$$
$$\leq C \left[ s^2 \left( \sqrt{\frac{\log d}{n}} + \frac{4r^2\sqrt{2\ln 1.25/\delta}\sqrt{\log d}}{\varepsilon\gamma n} \right)^{2-2q} + \left( \sqrt{\frac{\log d}{n}} + \frac{4r^2\sqrt{2\ln 1.25/\delta}\sqrt{\log d}}{\varepsilon\gamma n} \right)^2 \right]$$

$$\tag{16}$$

for some constant $C$. Since $\ddot{\Sigma}_{\bar{X}\bar{X}} - \Sigma$ is symmetric, we know that $\|\ddot{\Sigma}_{\bar{X}\bar{X}} - \Sigma\|_1 = \|\ddot{\Sigma}_{\bar{X}\bar{X}} - \Sigma\|_\infty$. Thus, it suffices to prove that the bound in equation 16 holds for $\|\ddot{\Sigma}_{\bar{X}\bar{X}} - \Sigma\|_1$.

Define the event $A_{ij}$ by

$$A_{ij} = \left\{ \left| \ddot{\sigma}_{\bar{x}\bar{x}^T,ij} - \sigma_{xx^T,ij} \right| \leq 4 \min \left\{ \left| \sigma_{xx^T,ij} \right|, Thres \right\} \right\}. \tag{17}$$

Then by lemma 25, we have $\mathbb{P}(A_{ij}) \geq 1 - 2C_1 d^{-9/2}$.

Let $D = (d_{ij})_{1 \leq i,j \leq d}$ with $d_{ij} = \left( \ddot{\sigma}_{\bar{x}\bar{x}^T,ij} - \sigma_{xx^T,ij} \right) I\left( A_{ij}^c \right)$, then the following holds.

$$\mathbb{E}\|\ddot{\Sigma}_{\bar{X}\bar{X}} - \Sigma\|_1^2 \leq 2\mathbb{E} \left[ \sup_j \sum_{i \neq j} \left| \ddot{\sigma}_{\bar{x}\bar{x}^T,ij} - \sigma_{xx^T,ij} \right| I\left(A_{ij}\right) \right]^2 + 2\mathbb{E}\|D\|_1^2$$
$$+ C \left( \sqrt{\frac{\log d}{n}} + \frac{4r^2\sqrt{2\ln 1.25/\delta}\sqrt{\log d}}{\varepsilon\gamma n} \right)^2$$
$$\leq 32 \left[ \sup_j \sum_{i \neq j} \min \left\{ \left| \sigma_{xx^T,ij} \right|, \gamma\sqrt{\frac{\log d}{n}} + \frac{4r^2\sqrt{2\ln 1.25/\delta}\sqrt{\log d}}{\varepsilon n} \right\} \right]^2 + 2\mathbb{E}\|D\|_1^2$$
$$+ C \left( \sqrt{\frac{\log d}{n}} + \frac{4r^2\sqrt{2\ln 1.25/\delta}\sqrt{\log d}}{\varepsilon\gamma n} \right)^2.$$

$$\tag{18}$$

Note the first term in equation 18 is bounded by $Cs^2 \left( \sqrt{\frac{\log d}{n}} + \frac{4r^2\sqrt{2\ln 1.25/\delta}\sqrt{\log d}}{\varepsilon\gamma n} \right)^{2-2q}$, and the first term is dominating, while the second term $\mathbb{E}\|D\|_1^2$ is comparably negligible. By setting

$$k^* = \left\lfloor s \left( \sqrt{\frac{\log d}{n}} + \frac{4r^2\sqrt{2\ln 1.25/\delta}\sqrt{\log d}}{\varepsilon \gamma n} \right)^{-q} \right\rfloor, \text{ we have:}$$

$$\sum_{i \neq j} \min \left\{ \left| \sigma_{xx^T,ij} \right|, \gamma\sqrt{\frac{\log d}{n}} + \frac{4r^2\sqrt{2\ln 1.25/\delta}\sqrt{\log d}}{\varepsilon n} \right\}$$

$$\leq \gamma \left( \sum_{i \leq k^*} + \sum_{i > k^*} \right) \min \left\{ \left| \sigma_{[i]j} \right|, \sqrt{\frac{\log d}{n}} + \frac{4r^2\sqrt{2\ln 1.25/\delta}\sqrt{\log d}}{\varepsilon \gamma n} \right\}$$

$$\leq C_5 k^* \left( \sqrt{\frac{\log d}{n}} + \frac{4r^2\sqrt{2\ln 1.25/\delta}\sqrt{\log d}}{\varepsilon \gamma n} \right) + C_5 \sum_{i > k^*} \left( \frac{s}{i} \right)^{1/q}$$

$$\leq C_6 \left[ k^* \left( \sqrt{\frac{\log d}{n}} + \frac{4r^2\sqrt{2\ln 1.25/\delta}\sqrt{\log d}}{\varepsilon \gamma n} \right) + s^{1/q} \cdot (k^*)^{1-1/q} \right]$$

$$\leq C_7 s \left( \sqrt{\frac{\log d}{n}} + \frac{4r^2\sqrt{2\ln 1.25/\delta}\sqrt{\log d}}{\varepsilon \gamma n} \right)^{1-q}$$

which implies equation 16 if $\mathbb{E}\|D\|_1^2 = O\left(\frac{1}{n}\right)$. We shall now show that $\mathbb{E}\|D\|_1^2 = O\left(\frac{1}{n}\right)$. Note that:

$$\mathbb{E}\|D\|_1^2 \leq d \sum_{ij} \mathbb{E} d_{ij}^2$$

$$= d \sum_{ij} \mathbb{E} \left\{ \left[ d_{ij}^2 I \left( A_{ij}^c \cap \{\ddot{\sigma}_{\bar{x}\bar{x}^T,ij} = \dot{\sigma}_{\bar{x}\bar{x}^T,ij}\} \right) + d_{ij}^2 I \left( A_{ij}^c \cap \{\ddot{\sigma}_{\bar{x}\bar{x}^T,ij} = 0\} \right) \right] \right\}$$

$$= d \sum_{ij} \mathbb{E} \left\{ \left( \dot{\sigma}_{\bar{x}\bar{x}^T,ij} - \sigma_{xx^T,ij} \right)^2 I \left( A_{ij}^c \right) \right\} + d \sum_{ij} \mathbb{E} \sigma_{xx^T,ij}^2 I \left( A_{ij}^c \cap \{\ddot{\sigma}_{\bar{x}\bar{x}^T,ij} = 0\} \right)$$

$$\equiv R_1 + R_2$$

Lemma 25 yields that $\mathbb{P}\left(A_{ij}^c\right) \leq 2C_1 d^{-9/2}$, and the Whittle inequality (Theorem 2 in [56]) implies $\dot{\sigma}_{\bar{x}\bar{x}^T,ij} - \sigma_{xx^T,ij}$ has all finite moments under the sub-Gaussianity condition. Hence, we have:

$$R_1 = d \sum_{ij} \mathbb{E} \left\{ \left( \dot{\sigma}_{\bar{x}\bar{x}^T,ij} - \sigma_{xx^T,ij} \right)^2 I \left( A_{ij}^c \right) \right\}$$

$$\leq d \sum_{ij} \left[ \mathbb{E} \left( \dot{\sigma}_{\bar{x}\bar{x}^T,ij} - \sigma_{xx^T,ij} \right)^6 \right]^{1/3} \mathbb{P}^{2/3} \left( A_{ij}^c \right)$$

$$\leq C_8 d \cdot d^2 \cdot \frac{1}{n} \cdot d^{-3} = C_8/n$$

On the other hand,

$$R_2 = d \sum_{ij} \mathbb{E} \sigma_{xx^T,ij}^2 I \left( A_{ij}^c \cap \{\ddot{\sigma}_{\bar{x}\bar{x}^T,ij} = 0\} \right)$$

$$= d \sum_{ij} \mathbb{E} \sigma_{xx^T,ij}^2 I \left( \left| \sigma_{xx^T,ij} \right| \geq 4 \left( \gamma\sqrt{\frac{\log d}{n}} + \frac{4r^2\sqrt{2\ln 1.25/\delta}\sqrt{\log d}}{\varepsilon} \right) \right)$$

$$I \left( \left| \dot{\sigma}_{\bar{x}\bar{x}^T,ij} \right| \leq \gamma\sqrt{\frac{\log d}{n}} + \frac{4r^2\sqrt{2\ln 1.25/\delta}\sqrt{\log d}}{\varepsilon} \right)$$

$$\leq d \sum_{ij} \sigma_{xx^T,ij}^2 \mathbb{E} I \left( \left| \sigma_{xx^T,ij} \right| 4\gamma\sqrt{\frac{\log d}{n}} + \frac{16r^2\sqrt{2\ln 1.25/\delta}\sqrt{\log d}}{\varepsilon} \right)$$

$$I \left( \left| \sigma_{xx^T,ij} \right| - \left| \dot{\sigma}_{\bar{x}\bar{x}^T,ij} - \sigma_{xx^T,ij} \right| \leq \gamma\sqrt{\frac{\log d}{n}} + \frac{16r^2\sqrt{2\ln 1.25/\delta}\sqrt{\log d}}{\varepsilon} \right)$$

$$\leq d \sum_{ij} \sigma^2_{xx^T,ij} \mathbb{E} I \left( \left| \dot{\sigma}_{\bar{x}\bar{x}^T,ij} - \sigma_{xx^T,ij} \right| > \frac{3}{4} \left| \sigma_{xx^T,ij} \right| \right)$$

$$I \left( \left| \sigma_{xx^T,ij} \right| \geq 4 \left( \gamma \sqrt{\frac{\log d}{n}} + \frac{16r^2 \sqrt{2 \ln 1.25/\delta} \sqrt{\log d}}{\varepsilon} \right) \right)$$

$$\leq d \sum_{ij} \sigma^2_{xx^T,ij} \mathbb{E} I \left( \left| \sigma_{xx^T,ij} \right| > 4\gamma \sqrt{\frac{\log p}{n}} + \frac{16r^2 \sqrt{2 \ln 1.25/\delta} \sqrt{\log p}}{n\varepsilon} \right)$$

$$I \left( \left| \hat{\sigma}_{\bar{x}\bar{x}^T,ij} - \sigma_{xx^T,ij} \right| + |n_{1,ij}| \geq \frac{3}{4} \left| \sigma_{xx^T,ij} \right| \right)$$

$$\leq d \sum_{ij} \sigma^2_{xx^T,ij} \mathbb{P} \left( \left\{ \left| \hat{\sigma}_{\bar{x}\bar{x}^T,ij} - \sigma_{xx^T,ij} \right| \geq \frac{3}{4} \left| \sigma_{xx^T,ij} \right| - |n_{1,ij}| \right\} \right.$$

$$\left. \bigcap \left\{ \left| \sigma_{xx^T,ij} \right| > 4\gamma \sqrt{\frac{\log d}{n}} + \frac{16r^2 \sqrt{2 \ln 1.25/\delta} \sqrt{\log d}}{n\varepsilon} \right\} \right)$$

$$= d \sum_{ij} \sigma^2_{xx^T,ij} \mathbb{P} \left( \left\{ \left| \hat{\sigma}_{\bar{x}\bar{x}^T,ij} - \sigma_{xx^T,ij} \right| \geq \frac{3}{4} \left| \sigma_{xx^T,ij} \right| - |n_{1,ij}| \right\} \bigcap \left\{ |n_{1,ij}| \leq \frac{1}{4} \left| \sigma_{xx^T,ij} \right| \right\} \right.$$

$$\left. \bigcap \left\{ \left| \sigma_{xx^T,ij} \right| > 4\gamma \sqrt{\frac{\log d}{n}} + \frac{16r^2 \sqrt{2 \ln 1.25/\delta} \sqrt{\log d}}{n\varepsilon} \right\} \right)$$

$$+ d \sum_{ij} \sigma^2_{xx^T,ij} \mathbb{P} \left( \left\{ \left| \hat{\sigma}_{\bar{x}\bar{x}^T,ij} - \sigma_{xx^T,ij} \right| \geq \frac{3}{4} \left| \sigma_{xx^T,ij} \right| - |n_{1,ij}| \right\} \right.$$

$$\left. \bigcap \left\{ |n_{1,ij}| \geq \frac{1}{4} \left| \sigma_{xx^T,ij} \right| \right\} \bigcap \left\{ \left| \sigma_{xx^T,ij} \right| > 4\gamma \sqrt{\frac{\log d}{n}} + \frac{16r^2 \sqrt{2 \ln 1.25/\delta} \sqrt{\log d}}{n\varepsilon} \right\} \right)$$

This gives us:

$$R_2 \leq d \sum_{ij} \sigma^2_{xx^T,ij} \mathbb{P} \left( \left\{ \left| \hat{\sigma}_{\bar{x}\bar{x}^T,ij} - \sigma_{xx^T,ij} \right| \geq \frac{1}{2} \left| \sigma_{xx^T,ij} \right| \right\} \right.$$

$$\bigcap \left\{ \left| \sigma_{xx^T,ij} \right| > 4\gamma \sqrt{\frac{\log d}{n}} + \frac{16r^2 \sqrt{2 \ln 1.25/\delta} \sqrt{\log d}}{n\varepsilon} \right\} \right)$$

$$+ d \sum_{ij} \sigma^2_{xx^T,ij} \mathbb{P} \left( \left\{ |n_{1,ij}| \geq \frac{1}{4} \left| \sigma_{xx^T,ij} \right| \right\} \right. \tag{19}$$

$$\left. \bigcap \left\{ \left| \sigma_{xx^T,ij} \right| > 4\gamma \sqrt{\frac{\log d}{n}} + \frac{16r^2 \sqrt{2 \ln 1.25/\delta} \sqrt{\log d}}{n\varepsilon} \right\} \right).$$

For the first term of equation 19, by Lemma 24 we have:

$$d \sum_{ij} \sigma^2_{xx^T,ij} \mathbb{P} \left( \left\{ \left| \hat{\sigma}_{\bar{x}\bar{x}^T,ij} - \sigma_{xx^T,ij} \right| \geq \frac{1}{2} \left| \sigma_{xx^T,ij} \right| \right\} \bigcap \left\{ \left| \sigma_{xx^T,ij} \right| \geq 4\gamma \sqrt{\frac{\log d}{n}} \right\} \right)$$

$$\leq \frac{d}{n} \sum_{ij} n\sigma^2_{xx^T,ij} \exp \left( -n \frac{2\sigma^2_{xx^T,ij}}{\gamma^2} \right) I \left( \left| \sigma_{xx^T,ij} \right| \geq 4\gamma \sqrt{\frac{\log d}{n}} \right)$$

$$\leq \frac{d}{n} \sum_{ij} \left[ n\sigma^2_{xx^T,ij} \exp \left( -n \frac{\sigma^2_{xx^T,ij}}{\gamma^2} \right) \right] \exp \left( -n \frac{\sigma^2_{xx^T,ij}}{\gamma^2} \right) I \left( \left| \sigma_{xx^T,ij} \right| \geq 4\gamma \sqrt{\frac{\log d}{n}} \right)$$

$$\leq C \frac{d^3}{n} d^{-16} = O \left( \frac{1}{n} \right)$$

For the second term of equation 19, by Lemmas 11 and 24 we have

$$d\sum_{ij}\sigma_{xx^T,ij}^2\mathbb{P}\left(\left\{|n_{1,ij}|\geq\frac{1}{4}\left|\sigma_{xx^T,ij}\right|\right\}\bigcap\left\{|\sigma_{xx^T,ij}|>4\gamma\sqrt{\frac{\log d}{n}}+\frac{16r^2\sqrt{2\ln 1.25/\delta}\sqrt{\log d}}{n\varepsilon}\right\}\right)$$

$$\leq d\sum_{ij}\sigma_{xx^T,ij}^2\mathbb{P}\left(|n_{1,ij}|\geq\gamma\sqrt{\frac{\log d}{n}}+\frac{4\sqrt{2\ln 1.25/\delta}\log d}{n\varepsilon}\right\}\right)\mathbb{P}\left(|n_{1,ij}|>\frac{1}{4}\sigma_{xx^T,ij}\right)$$

$$\leq Cd\sum_{ij}\sigma_{xx^T,ij}^2\cdot\exp\left(-\frac{\left(\gamma\sqrt{\frac{\log d}{n}}+4\sigma_1\sqrt{\log d}\right)^2}{2\sigma_1^2}\right)\exp\left(-\frac{\sigma_{xx^T,ij}^2}{32\sigma_1^2}\right)$$

$$\leq C\sigma_1^2 d\cdot d^2\exp\left(-\frac{\gamma^2\log d}{2n\sigma_1^2}\right)d^{-8}$$

$$\leq C\sigma_1^2 d^{-5}\frac{2n\sigma_1^2}{\gamma^2\log d}=O\left(\frac{\log 1/\delta}{n\varepsilon^2}\right).$$

Putting $R_1$ and $R_2$ together yields that for some constant $C>0$,

$$\mathbb{E}\|D\|_1^2\leq\frac{C}{n}.$$

$\square$

**Proof of Lemma. 25.** Firstly, let us note that $\dot{\Sigma}_{\bar{X}\bar{X}}=\hat{\Sigma}_{\bar{X}\bar{X}}+N_1=\sum_i^n\bar{x}_i\bar{x}_i^T+N_1$. Therefore by Lemma 24 and Lemma 11 with probability at least $1-Cd^{-8}$, for all $1\leq i,j\leq d$, and for some constants $\gamma$ and $C$ that depends on $\sigma_{N_1}$,

$$\left|\dot{\sigma}_{\bar{x}\bar{x}^T,ij}-\sigma_{xx^T,ij}\right|\leq\gamma\sqrt{\frac{\log d}{n}}+\frac{128r^2\sqrt{2\log\frac{1.25}{\delta}\log d}}{n\varepsilon}\leq O\left(\gamma r^2\frac{\sqrt{\log d\log\frac{1}{\delta}}}{\sqrt{n}\varepsilon}\right). \qquad (20)$$

Define the event $A_{ij}=\left\{\left|\dot{\sigma}_{\bar{x}\bar{x}^T,ij}\right|>\gamma\sqrt{\frac{\log d}{n}}+\frac{4r^2\sqrt{2\ln 1.25/\delta}\sqrt{\log d}}{\varepsilon n}\right\}$. Let $\hat{\Sigma}_{\bar{X}\bar{X}}=\left(\hat{\sigma}_{\bar{x}\bar{x}^T,ij}\right)_{1\leq i,j\leq d}$ and $N_1=(n_{1,ij})_{1\leq i,j\leq d}$. We have:

$$\left|\ddot{\sigma}_{\bar{x}\bar{x}^T,ij}-\sigma_{xx^T,ij}\right|=\left|\sigma_{xx^T,ij}\right|\cdot I\left(A_{ij}^c\right)+\left|\dot{\sigma}_{\bar{x}\bar{x}^T,ij}-\sigma_{xx^T,ij}\right|\cdot I\left(A_{ij}\right). \qquad (21)$$

By the triangle inequality, it is easy to see that

$$A_{ij}=\left\{\left|\dot{\sigma}_{\bar{x}\bar{x}^T,ij}-\sigma_{xx^T,ij}+\sigma_{xx^T,ij}\right|>\gamma\sqrt{\frac{\log d}{n}}+\frac{4r^2\sqrt{2\ln 1.25/\delta}\sqrt{\log d}}{\varepsilon n}\right\}$$

$$\subset\left\{\left|\dot{\sigma}_{\bar{x}\bar{x}^T,ij}-\sigma_{xx^T,ij}\right|>\gamma\sqrt{\frac{\log d}{n}}+\frac{4r^2\sqrt{2\ln 1.25/\delta}\sqrt{\log d}}{\varepsilon n}-\left|\sigma_{xx^T,ij}\right|\right\}$$

and

$$A_{ij}^c=\left\{\left|\dot{\sigma}_{\bar{x}\bar{x}^T,ij}-\sigma_{xx^T,ij}+\sigma_{xx^T,ij}\right|\leq\gamma\sqrt{\frac{\log d}{n}}+\frac{4r^2\sqrt{2\ln 1.25/\delta}\sqrt{\log d}}{\varepsilon n}\right\}$$

$$\subset\left\{\left|\dot{\sigma}_{\bar{x}\bar{x}^T,ij}-\sigma_{xx^T,ij}\right|>\left|\sigma_{xx^T,ij}\right|-\left(\gamma\sqrt{\frac{\log d}{n}}+\frac{4r^2\sqrt{2\ln 1.25/\delta}\sqrt{\log d}}{\varepsilon n}\right)\right\}.$$

Depending on the value of $\sigma_{xx^T,ij}$, we need to consider the following three cases.

**Case 1.** $\left|\sigma_{xx^T,ij}\right|\leq\frac{\gamma}{4}\sqrt{\frac{\log d}{n}}+\frac{\sqrt{2\log 1.25/\delta}\sqrt{\log d}}{n\varepsilon}$.

For this case, we have:

$$\mathbb{P}\left(A_{ij}\right) \leq \mathbb{P}\left(\left|\dot{\sigma}_{\bar{x}\bar{x}^T,ij} - \sigma_{xx^T,ij}\right| > \frac{3\gamma}{4}\sqrt{\frac{\log d}{n}} + \frac{3\sqrt{2\ln 1.25/\delta}\sqrt{\log d}}{n\varepsilon}\right) \leq C_1 d^{-\frac{9}{2}} + 2d^{-\frac{9}{2}}.$$

This is due to the fact:

$$\mathbb{P}\left(\left|\dot{\sigma}_{\bar{x}\bar{x}^T,ij} - \sigma_{xx^T,ij}\right| > \frac{3\gamma}{4}\sqrt{\frac{\log d}{n}} + \frac{3\sqrt{2\ln 1.25/\delta}\sqrt{\log d}}{n\varepsilon}\right)$$

$$\leq \mathbb{P}\left(\left|\hat{\sigma}_{\bar{x}\bar{x}^T,ij} - \sigma_{xx^T,ij}\right| > \frac{3\gamma}{4}\sqrt{\frac{\log d}{n}} + \frac{3\sqrt{2\ln 1.25/\delta}\sqrt{\log d}}{n\varepsilon}\right) - |n_{1,ij}|\right)$$

$$= \mathbb{P}\left(B_{ij}\bigcap\left\{\frac{3\sqrt{2\ln 1.25/\delta}\sqrt{\log d}}{n\varepsilon}\right) - |n_{1,ij}| > 0\right\}\right)$$

$$+\mathbb{P}\left(B_{ij}\bigcap\left\{\frac{3\sqrt{2\ln 1.25/\delta}\sqrt{\log d}}{n\varepsilon}\right) - |n_{1,ij}| \leq 0\right\}\right)$$

$$\leq \mathbb{P}\left(\left|\hat{\sigma}_{\bar{x}\bar{x}^T,ij} - \sigma_{xx^T,ij}\right| > \frac{3\gamma}{4}\sqrt{\frac{\log d}{n}}\right) + \mathbb{P}\left(\frac{2\sqrt{3\ln 1.25/\delta}\log d}{n\varepsilon}\right) \leq |n_{1,ij}|\right)$$

$$\leq C_1 d^{-\frac{9}{2}} + 2d^{-\frac{9}{2}},$$

where event $B_{ij}$ denotes $B_{ij} = \left\{\left|\hat{\sigma}_{\bar{x}\bar{x}^T,ij} - \sigma_{xx^T,ij}\right| > \frac{3\gamma}{4}\sqrt{\frac{\log d}{n}} + \frac{2\sqrt{2\ln 1.25/\delta}\log d}{n\varepsilon}\right) - |n_{1,ij}|\right\}$, and the last inequality comes from lemma 11 and 24. Thus by equation 21, with probability at least $1 - C_1 d^{-\frac{9}{2}} - 2d^{-\frac{9}{2}}$, we have: $\left|\ddot{\sigma}_{\bar{x}\bar{x}^T,ij} - \sigma_{xx^T,ij}\right| = \left|\sigma_{xx^T,ij}\right|$, which satisfies equation 5.

**Case 2.** $\left|\sigma_{xx^T,ij}\right| \geq 2\gamma\sqrt{\frac{\log d}{n}} + \frac{8r^2\sqrt{2\ln 1.25/\delta}\sqrt{\log d}}{n\varepsilon}$.

For this case, we have

$$\mathbb{P}\left(A_{ij}^c\right) \leq \mathbb{P}\left(\left|\dot{\sigma}_{\bar{x}\bar{x}^T,ij} - \sigma_{xx^T,ij}\right| \geq \gamma\sqrt{\frac{\log d}{n}} + \frac{r^2\sqrt{2\ln 1.25/\delta}\sqrt{\log d}}{\varepsilon n}\right) \leq C_1 d^{-8} + 2d^{-8},$$

where the proof is identical to the proof for case 1. Thus, with probability at least $1 - C_1 d^{-\frac{9}{2}} - 2d^{-8}$, we have:

$$\left|\ddot{\sigma}_{\bar{x}\bar{x}^T,ij} - \sigma_{xx^T,ij}\right| = \left|\dot{\sigma}_{\bar{x}\bar{x}^T,ij} - \sigma_{xx^T,ij}\right|.$$

Also, by equation 20, equation 5 also holds.

**Case 3.** Otherwise,

$$\frac{\gamma}{4}\sqrt{\frac{\log d}{n}} + \frac{r^2\sqrt{2\log 1.25/\delta}\sqrt{\log d}}{n\varepsilon} \leq \left|\sigma_{xx^T,ij}\right| \leq 2\gamma\sqrt{\frac{\log d}{n}} + \frac{8r^2\sqrt{2\ln 1.25/\delta}\sqrt{\log d}}{n\varepsilon}.$$

For this case, we have

$$\left|\ddot{\sigma}_{\bar{x}\bar{x}^T,ij} - \sigma_{xx^T,ij}\right| = \left|\sigma_{xx^T,ij}\right| \text{ or } \left|\dot{\sigma}_{\bar{x}\bar{x}^T,ij} - \sigma_{xx^T,ij}\right|.$$

When $\left|\sigma_{xx^T,ij}\right| \leq \gamma\sqrt{\frac{\log d}{n}} + \frac{4r^2\sqrt{2\ln 1.25/\delta}\sqrt{\log d}}{\varepsilon n}$, we can derive from equation 20 that with probability at least $1 - 2d^{-6} - C_1 d^{-8}$

$$\left|\dot{\sigma}_{\bar{x}\bar{x}^T,ij} - \sigma_{xx^T,ij}\right| \leq \gamma\sqrt{\frac{\log d}{n}} + \frac{4r^2\sqrt{2\ln 1.25/\delta}\sqrt{\log d}}{\varepsilon n} \leq 4\left|\sigma_{xx^T,ij}\right|.$$

Thus, equation 5 holds whether $\left|\sigma_{xx^T,ij}\right| \geq \gamma\sqrt{\frac{\log d}{n}} + \frac{4r^2\sqrt{2\ln 1.25/\delta}\sqrt{\log d}}{\varepsilon n}$ or not, which completes the proof of Lemma 25. $\qquad\square$

# G Omitted Proofs

**Proof of Theorem 2.** By Gaussian mechanism and the post-processing processing property, it is easily to see that releasing $\ddot{\Sigma}_{\bar{X}\bar{X}}$ satisfies $(\frac{\varepsilon}{2}, \frac{\delta}{2})$-DP, releasing $\dot{\Sigma}_{\widetilde{X}\widetilde{Y}}$ satisfies $(\frac{\varepsilon}{2}, \frac{\delta}{2})$-DP. Thus, the output of Algorithm 1 is $(\varepsilon, \delta)$-DP.

Next, we show that the output of the mechanism satisfies joint differential privacy using Billboard Lemma (Lemma 17). The estimators $\hat{\theta}^P(\hat{D}^0)$ and $\hat{\theta}^P(\hat{D}^1)$ are computed in the same way as $\hat{\theta}^P(\hat{D})$, so $\hat{\theta}^P(\hat{D}^0)$ and $\hat{\theta}^P(\hat{D}^1)$ each satisfy $(\varepsilon, \delta)$-JDP. Since $\hat{\theta}^P(\hat{D}^0)$ and $\hat{\theta}^P(\hat{D}^1)$ are computed on disjoint subsets of the data, then by the Parallel Composition Theorem, together they satisfy $(\varepsilon, 2\delta)$-JDP. By the Sequential Composition Theorem (Lemma 19), the estimators $(\hat{\theta}^P(\hat{D}), \hat{\theta}^P(\hat{D}^0), \hat{\theta}^P(\hat{D}^1))$ together satisfy $(2\varepsilon, 3\delta)$-JDP. Finally, using the post-processing property and Billboard Lemma 17, the output $(\bar{\theta}^P(\hat{D}), \bar{\theta}^P(\hat{D}^0), \bar{\theta}^P(\hat{D}^1), \{\pi_i(D_i, \bar{\theta}^P(\hat{D}^b))\}_{i=1}^n)$ of Algorithm 2 satisfies $(2\varepsilon, 3\delta)$-JDP.

$\square$

**Proof of Theorem 3.** For any realization $D$ held by agents, let $\hat{D} = \sigma_{\tau_{\alpha,\beta}}(D)$. Then by Lemma 15 we have

$$\mathbb{E}[\|\bar{\theta}^P(\hat{D}) - \theta^*\|_2^2]$$
$$\leq \mathbb{E}\|\hat{\theta}^P(\hat{D}) - \theta^*\|_2^2$$
$$= \mathbb{E}\|\hat{\theta}^P(\hat{D}) - \hat{\theta}^P(D) + \hat{\theta}^P(D) - \theta^*\|_2^2$$
$$= \mathbb{E}\|\hat{\theta}^P(\hat{D}) - \hat{\theta}^P(D)\|_2^2 + 2\langle \hat{\theta}^P(\hat{D}) - \hat{\theta}^P(D), \hat{\theta}^P(D) - \theta^* \rangle + \|\hat{\theta}^P(D) - \theta^*\|_2^2$$
$$\leq 2\mathbb{E}\|\hat{\theta}^P(\hat{D}) - \hat{\theta}^P(D)\|_2^2 + 2\mathbb{E}\|\hat{\theta}^P(D) - \theta^*\|_2^2. \tag{22}$$

For the first term of equation 22, if we set the constraint bound $\lambda_n = O\left(\frac{r^2\sqrt{\log d \log \frac{1}{\delta}}}{\sqrt{n}\varepsilon}\right)$, we know from Lemma 4 and Lemma 20 that when n is sufficient large such that $n \geq \Omega(\frac{s^2 r^4 \log d \log \frac{1}{\delta}}{\varepsilon^2 \kappa_\infty})$, with probability at least $1 - \beta - O(d^{-\Omega(1)})$ we have

$$\mathbb{E}\|\hat{\theta}^P(\hat{D}) - \hat{\theta}^P(D)\|_2 \leq 16\sqrt{k}\lambda_n \alpha n \tag{23}$$

For the last term of equation 22, by Lemma 22, when n is sufficient large such that $n \geq \Omega(\frac{s^2 r^4 \log d \log \frac{1}{\delta}}{\varepsilon^2 \kappa_\infty})$, with probability at least $1 - O(d^{-\Omega(1)})$,

$$\mathbb{E}\|\hat{\theta}(D) - \theta^*\|_2 \leq \sqrt{k}\lambda_n. \tag{24}$$

Combining equation 23 and equation 24 yields that with probability at least $1 - \beta - O(d^{-\Omega(1)})$,

$$\mathbb{E}[\|\bar{\theta}^P(\hat{D}) - \theta^*\|_2^2] \leq \mathbb{E}[\|\hat{\theta}^P(\hat{D}) - \theta^*\|_2^2] \leq O\left(k\alpha^2 \frac{r^4 \log d \log \frac{1}{\delta}}{\varepsilon^2} + \frac{r^4 \log d \log \frac{1}{\delta}}{n^2 \varepsilon^2}\right).$$

$\square$

**Proof of Theorem 5.** Suppose all agents other than $i$ are following strategy $\sigma_{\tau_{\alpha,\beta}}$. Let agent $i$ be in group $1 - b, b \in \{0, 1\}$. We will show that $\sigma_{\tau_{\alpha,\beta}}$ achieves $\eta$-Bayesian Nash equilibrium by bounding agent $i$'s incentive to deviate. Assume that $c_i \leq \tau_{\alpha,\beta}$, otherwise there is nothing to show because agent $i$ would be allowed to submit an arbitrary report under $\sigma_{\tau_{\alpha,\beta}}$. For ease of notation, we write $\sigma$ for $\sigma_{\tau_{\alpha,\beta}}$ for the remainder of the proof. We first compute the maximum expected amount (based on their belief) that agent $i$ can increase their payment by misreporting to the analyst, i.e.

$$\mathbb{E}\left[\pi_i(\hat{D}_i, \sigma(D^b, c^b))|D_i, c_i\right] - \mathbb{E}[\pi_i(D_i, \sigma(D^b, c^b))|D_i, c_i]$$
$$= \mathbb{E}\left[B_{a_1,a_2}\left(\langle x_i, \bar{\theta}^P(\hat{D}^b)\rangle), \langle \bar{x}_i, \mathbb{E}_{\theta \sim p(\theta|\hat{D}_i)}[\theta]\rangle\right) \Big| D_i, c_i\right]$$
$$- \mathbb{E}\left[B_{a_1,a_2}\left(\langle x_i, \bar{\theta}^P(\hat{D}^b)\rangle, \langle \bar{x}_i, \mathbb{E}_{\theta \sim p(\theta|D_i)}[\theta]\rangle\right) \Big| D_i, c_i\right]. \tag{25}$$

Note that $B_{a_1,a_2}(p,q) = a_1 - a_2(p - 2pq + q^2)$ is linear with respect to $p$, and is a strictly concave function of $q$ maximized at $q = p$. Thus, equation 25 is upper bounded by the following with probability $1 - C_1 n^{-\Omega(1)}$

$$B_{a_1,a_2}\left[\mathbb{E}\left[\langle \bar{x}_i, \bar{\theta}^P(\hat{D}^b)\rangle | D_i, c_i\right], \mathbb{E}\left[\langle \bar{x}_i, \hat{\theta}^P(\hat{D}^b)\rangle | D_i, c_i\right]\right]$$

$$- B_{a_1,a_2}\left[\mathbb{E}[\langle x_i, \bar{\theta}^P(\hat{D}^b)\rangle | D_i, c_i], \langle \bar{x}_i, \mathbb{E}_{\theta \sim p(\theta|D_i)}[\theta]\rangle\right]$$

$$= a_2\left(\mathbb{E}[\langle \bar{x}_i, \bar{\theta}^P(\hat{D}^b)\rangle | D_i, c_i] - \langle \bar{x}_i, \mathbb{E}_{\theta \sim p(\theta|D_i)}[\theta]\rangle\right)^2$$

$$= a_2\left(\mathbb{E}[\langle \bar{x}_i, \bar{\theta}^P(\hat{D}^b)\rangle - \langle \bar{x}_i, \mathbb{E}_{\theta \sim p(\theta|D_i)}[\theta]\rangle | D_i, c_i]\right)^2$$

$$\leq a_2\left(\mathbb{E}[\bar{x}_i^T(\bar{\theta}^P(\hat{D}^b) - \mathbb{E}_{\theta \sim p(\theta|D_i)}[\theta])| D_i, c_i]\right)^2$$

$$\leq a_2\|\bar{x}_i\|_2^2\|\mathbb{E}[\bar{\theta}^P(\hat{D}^b) - \mathbb{E}_{\theta \sim p(\theta|D_i)}[\theta]| D_i, c_i]\|_2^2$$

$$\leq r^2 a_2\|\mathbb{E}[\bar{\theta}^P(\hat{D}^b) - \mathbb{E}_{\theta \sim p(\theta|D_i)}[\theta]| D_i, c_i]\|_2^2.$$

We continue by bounding the term $\|\mathbb{E}[\bar{\theta}^P(\hat{D}^b) - \mathbb{E}_{\theta \sim p(\theta|D_i)}[\theta]| D_i, c_i]\|_2$. By Lemma 15

$$\|\mathbb{E}[\bar{\theta}^P(\hat{D}^b) - \mathbb{E}_{\theta \sim p(\theta|D_i)}[\theta]| D_i, c_i]\|_2$$

$$\leq \|\mathbb{E}[\bar{\theta}^P(\hat{D}^b) - \bar{\theta}^P(D^b)| D_i, c_i]\|_2 + \|\mathbb{E}[\bar{\theta}^P(D^b)| D_i, c_i] - \mathbb{E}_{\theta \sim p(\theta|D_i)}[\theta]| D_i, c_i]\|_2$$

$$\leq \|\mathbb{E}[\hat{\theta}^P(\hat{D}^b) - \hat{\theta}^P(D^b)| D_i, c_i]\|_2 + \|\mathbb{E}[\bar{\theta}^P(D^b)| D_i, c_i] - \mathbb{E}_{\theta \sim p(\theta|D_i)}[\theta]| D_i, c_i]\|_2$$

$$\leq \|\hat{\theta}^P(\hat{D}^b) - \hat{\theta}^P(D^b)\|_2 + \|\mathbb{E}[\bar{\theta}^P(D^b)| D_i] - \mathbb{E}_{\theta \sim p(\theta|D_i)}[\theta]\|_2 \tag{26}$$

For the first term of equation 26, since agent $i$ believes that with at least probability $1 - \beta$, at most $\alpha n$ agents will misreport their datasets under threshold strategy $\sigma_{\tau_{\alpha,\beta}}$, datasets $D^b$ and $\hat{D}^b$ differ only on at most $\alpha n$ agents' datasets. By Lemma 20 and Lemma 4, we set the constraint bound $\lambda_n = O\left(\frac{r^2\sqrt{\log d \log \frac{1}{\delta}}}{\sqrt{n}\varepsilon}\right)$, when n is sufficient large such that $n \geq \Omega(\frac{s^2 r^4 \log d \log \frac{1}{\delta}}{\varepsilon^2 \kappa_\infty})$, with probability at least $1 - \beta - O(\alpha n d^{-\Omega(1)})$ we have that

$$\mathbb{E}\|\hat{\theta}^P(\hat{D}) - \hat{\theta}^P(D)\|_2 \leq 16\sqrt{k}\lambda_n \tag{27}$$

For the second term of equation 26 :

$$\mathbb{E}[\bar{\theta}^P(D^b)|D_i] - \mathbb{E}_{\theta \sim p(\theta|D_i)}[\theta] = \mathbb{E}_{D^b \sim p(D^b|D_i)}[\bar{\theta}^P(D^b)] - \mathbb{E}_{\theta \sim p(\theta|D_i)}[\theta]$$

$$= \mathbb{E}_{\theta \sim p(\theta|D_i)}[\mathbb{E}_{D^b \sim p(D^b|\theta)}[\bar{\theta}^P(D^b)]|\theta] - \mathbb{E}_{\theta \sim p(\theta|D_i)}[\theta]$$

$$= \mathbb{E}_{\theta \sim p(\theta|D_i)}[\mathbb{E}_{D^b \sim p(D^b|\theta)}[\bar{\theta}^P(D^b) - \theta]|\theta].$$

Since

$$p(D^b|\theta) = p(X^b, y^b|\theta) = p(y^b|X^b, \theta)p(X^b|\theta) = p(y^b|X^b, \theta)p(X^b),$$

we have

$$\mathbb{E}_{D^b \sim p(D^b|\theta)}[\hat{\theta}(D^b) - \theta] = \mathbb{E}_{X^b}[\mathbb{E}_{y^b}[\bar{\theta}^P(X^b, y^b) - \theta]|X^b, \theta].$$

Since we have the prior knowledge that $\|\theta^*\|_2 \leq \tau_\theta$. Thus, for the posterior distribution $\theta \sim p(\theta|\hat{D}_i)$ it will also have $\|\theta\|_2 \leq \tau_\theta$. By Jensen's inequality, Theorem 3 and Lemma 15, we have

$$\|\mathbb{E}[\bar{\theta}^P(D^b)|D_i] - \mathbb{E}_{\theta \sim p(\theta|D_i)}[\theta]\|_2 \leq \mathbb{E}_{\theta \sim p(\theta|D_i), X^b}[\mathbb{E}_{y^b}[\|\bar{\theta}^P(X^b, y^b) - \theta\|_2|X^b, \theta]]$$

$$\leq \mathbb{E}_{\theta \sim p(\theta|D_i), X^b}[\mathbb{E}_{y^b}[\|\hat{\theta}^P(X^b, y^b) - \theta\|_2|X^b, \theta]]$$

$$\leq O\left(\frac{r^2\sqrt{k \log d \log \frac{1}{\delta}}}{n\varepsilon}\right)$$

In addition to an increased payment, agent $i$ may also experience decreased privacy costs from misreporting. By Assumption 3, this decrease in privacy costs is bounded above by $c_i 8(1 + 3\delta)\varepsilon^3$. Since we have assumed $c_i \leq \tau_{\alpha,\beta}$, the decrease in privacy costs for agent $i$ is bounded above by $\tau_{\alpha,\beta} 8(1 + 3\delta)\varepsilon^3$. Hence, agent $i$'s total incentive to deviate is bounded above by

$$\eta = O\left(a_2\left(\frac{\alpha^2 r^6 k \log d \log \frac{1}{\delta}}{\varepsilon^2} + \frac{r^4 \log d \log \frac{1}{\delta}}{n^2 \varepsilon^2}\right) + \tau_{\alpha,\beta}\delta\varepsilon^2\right).$$

$\square$

**Proof of Theorem 6.** Let agent $i$ have privacy cost $c_i \leq \tau_{\alpha,\beta}$ and consider agent $i$'s utility from participating in the mechanism. Suppose agent $i$ is in group $1 - b$, then their expected utility is

$$\mathbb{E}[u_i] = \mathbb{E}\left[B_{a_1,a_2}\left(\langle \bar{x}_i, \bar{\theta}^P(\hat{D}^b)\rangle, \langle \bar{x}_i, \mathbb{E}_{\theta \sim p(\theta|\hat{D}_i)}[\theta]\rangle\right) | D_i, c_i\right] - f_i(c_i, \varepsilon)$$

$$\geq B_{a_1,a_2}\left(\mathbb{E}(\langle \bar{x}_i, \bar{\theta}^P(\hat{D}^b)\rangle) | D_i, c_i, \langle \bar{x}_i, \mathbb{E}_{\theta \sim p(\theta|\hat{D}_i)}[\theta]\rangle\right) - \tau_{\alpha,\beta} 8(1 + 3\delta)\varepsilon^3. \quad (28)$$

Note that

$$B_{a_1,a_2}(p, q) = a_1 - a_2(p - 2pq + q^2) \geq a_1 - a_2(|p| + 2|p||q| + |q|^2), \quad (29)$$

Since both $|\langle \bar{x}_i, \bar{\theta}^P(\hat{D}^b)\rangle|$ and $|\langle \bar{x}_i, \mathbb{E}_{\theta \sim p(\theta|\hat{D}_i)}[\theta]\rangle|$ are bounded by $\|\bar{x}_i\|_2 \|\hat{\theta}(\hat{D}^b)\|_2 \leq r\tau_\theta$, thus by equation 28 and equation 29 agent $i$'s expected utility is non-negative as long as

$$a_1 \geq a_2(r\tau_\theta + 3r^2\tau_\theta^2) + \tau_{\alpha,\beta} 8(1 + 3\delta)\varepsilon^3.$$

$\square$

**Proof of Theorem 7.** Note that

$$B_{a_1,a_2}(p, q) \leq B_{a_1,a_2}(p, p) = a_1 - a_2(p - p^2) \leq a_1 + a_2(|p| + |p|^2),$$

thus

$$\mathcal{B} = \sum_{i=1}^n \mathbb{E}[\pi_i] = \sum_{i=1}^n \mathbb{E}[B_{a_1,a_2}\left(\langle \bar{x}_i, \bar{\theta}^P(\hat{D}^b)\rangle, \langle \bar{x}_i, \mathbb{E}_{\theta \sim p(\theta|\hat{D}_i)}[\theta]\rangle\right) | D_i, c_i]$$

$$\leq n(a_1 + a_2(r\tau_\theta + r^2\tau_\theta^2)).$$

$\square$

**Proof of Corollary 8.** For any $\xi \in (\frac{1}{3}, \frac{1}{2})$ and $c > 0$, we set

$$\varepsilon = n^{-\xi}, \delta = n^{-\Omega(1)},$$

$$\alpha = \Theta(n^{-3\xi}), \beta = \Theta(n^{-c}),$$

$$a_1 = a_2(r\tau_\theta + 3r^2\tau_\theta^2) + \tau_{\alpha,\beta} 8(1 + 3\delta)\varepsilon^3, a_2 = O(n^{-3\xi}).$$

Note that by choosing $\xi \in (\frac{1}{3}, \frac{1}{2})$, we ensure that $\alpha n = o(1)$.

Recall that by Theorem 3, the private estimator is $O\left(kn\alpha^2 \frac{r^4 \log d \log \frac{1}{\xi}}{\varepsilon^2} + \frac{r^4 \log d \log \frac{1}{\delta}}{n\varepsilon^2}\right)$-accurate.

Note that $O(\frac{n\alpha^2}{\varepsilon^2}) = O(n^{1-4\xi})$, $O(\frac{1}{n\varepsilon^2}) = O(n^{2\xi-1})$. Since for any $\xi \in (\frac{1}{3}, \frac{1}{2})$, we always have $1 - 4\xi < 2\xi - 1$, we obtain $\mathbb{E}\|\bar{\theta}^P(\hat{D}) - \theta^*\|_2^2 = O(n^{2\xi-1})$.

To bound the expected budget and truthfulness, we first consider bounding the threshold value $\tau_{\alpha,\beta}$ and the term $8(1 + 3\delta)\varepsilon^3$. By Lemma 14, $\tau_{\alpha,\beta} \leq \frac{1}{\lambda}\log\frac{1}{\alpha\beta} = \Theta(\frac{3\xi+c}{\lambda}\log n) = \widetilde{\Theta}(1)$. Combining these, we get $\tau_{\alpha,\beta} 8(1 + 3\delta)\varepsilon^3 = O(n^{-3\xi})$.

Now we bound the term $\eta$ for the truthfulness. Recall that by Theorem 5, the first term of the truthfulness bound is $a_2\left(\frac{n\alpha^2 r^6 k \log d \log \frac{1}{\delta}}{\varepsilon^2} + \frac{r^4 \log d \log \frac{1}{\delta}}{n\varepsilon^2}\right) = O(n^{-1-\xi})$, and the second term of the bound is $\tau_{\alpha,\beta} 8(1 + 3\delta)\varepsilon^3 = O(n^{-3\xi})$, thus $\eta = O(n^{-1-\xi} + n^{-3\xi}) = O(n^{-1-\xi})$.

Then we consider individual rationality. By the choice of $a_1$ and Theorem 6, the mechanism is individual rational for at least $1 - O(n^{-3\xi})$ fraction of agents.

Lastly, we consider total payment made to the agents. By Theorem 7, the total expected budget is $\mathcal{B} = O\left(n(a_1 + a_2(r\tau_\theta + r^2\tau_\theta^2))\right) = O(n(a_2 + \varepsilon^3 + a_2)) = O(n^{1-3\xi})$. $\qquad\square$

## H  Impact Statement

This paper presents work whose goal is to advance the field of machine learning theory. There are many potential societal consequences of our work, none which we feel must be specifically highlighted here.

