# OpenReview forum: "Truthful High Dimensional Sparse Linear Regression"
_NeurIPS.cc/2024/Conference — NeurIPS 2024 poster_

### Official Review · Reviewer_81kj · 2024-06-27

**Soundness:** 2
**Presentation:** 2
**Contribution:** 4
**Rating:** 5
**Confidence:** 2

**Summary:**

The authors present an $\varepsilon$-Bayesian incentive compatible and individually rational $k$-sparse linear regression algorithm with side payments for (almost all) privacy-oriented data providing agents. Remarkably, in the limit $d >> n >> \log d$, the accuracy and the needed budget both vanish.

**Strengths:**

The paper tackles an interesting problem, at the heart of data collection for machine learning applications.

The algorithms and the proof techniques are important advancements in the understanding of privacy-preserving machine learning with financial compensations to strategic data providers.

**Weaknesses:**

The paper already has a lot of different notations, which are easily confusing for the reader. It is thus very important to significantly improve the writing, which is too often extremely hard on the reader:
- Isn't the $\Pi_{\tau_\theta}$ the same operator as clipping (lines 2 and 3 of Algorithm 1)? It would make it much easier to read if all identical operators were written in the same way.
- The partition $\hat D^0 \cup \hat D^1$ should be presented before its use in line 273.
- Assumption 5 has a typo (I guess). It should be $D-i$ rather than $D_j$. There should also be a quantification over $i$ (unless each (c_i, D_i) is iid?).
- The "symmetric threshold strategy" is used in the main text without definition (and without even a reference to the appendix!).

The paper makes a common ground-truth assumption (same $\theta^*$). In practice, this may be unrealistic.

**Questions:**

I don't see Assumption 3 as a generalization of Cummings et al. Given that $\delta \leq 1$, the assumption is essentially equivalent to (at least implied by) $f_i \leq 2c_i \varepsilon^3$, which is more demanding than $f_i \leq c_i \varepsilon^2$. It should be rather presented as a relaxation. Do the authors agree?

I don't understand the claim (line 162) that "the one-round communication necessitates a closed-form estimator". Wouldn't any algorithm work? If so, I suggest that the authors merely present their new estimator as a proposal for covariance estimation, with a high-probability inversion guarantee (and even point to Theorem 3 as the end goal).

I fail to understand how $\theta^*$ can be recovered without assumptions on the thresholds $r$, $\tau_x$, $\tau_y$ and $\lambda_n$. It seems that this should not be possible for arbitrary values of these parameters. Can the authors clarify this?

Right now, though I think I mostly understand the algorithm, and its implications in terms of privacy, strategyproofness and individual rationality, its accuracy is a mystery to me, which prevents me from providing a larger score. I would be greatly thankful to the authors if they could help me gain insights into this aspect of their result.

**Limitations:**

The authors should better stress that
(i) Agents' "rationality" is merely on privacy leakage (they are indifferent to the trained model's behavior).
(ii) Agents are assumed to have the same labeling function (parameter $\theta^*$).
(iii) (Individually rational) agents whose data is too different are more likely removed from the system.

---

> ### Author Rebuttal · Authors · 2024-08-06
>
> >**W1: Isn't the $\Pi_{\tau_{\theta}}$ the same operator as clipping (lines 2 and 3 of Algorithm 1)? It would make it much easier to read if all identical operators were written in the same way.**
>
> We wish to thank the reviewer for pointing out this. We will add $\Pi_r$ to Line 2 of Algorithm 1  and unify the rest of the notations.
>
> >**W2: The partition $\hat{D}^{0} \cap \hat{D}^1$ should be presented before its use in line 273.**
>
>  Yes, we will notate it before its use in the finalized version.
>
> >**W3: Assumption 5 has a typo (I guess). It should be $D_{-i}$ rather than $D_{j}$.  There should also be a quantification over $i$
>  (unless each $(c_i, D_i)$ is iid?).**
>
> Yes, there is a typo: $D_j$ should be $D_i$. i.e., the conditional marginal distribution of $c_i$.  We are indeed taking the infimum over every data $D_i$. This is not supposed to exclude user $i$. And yes, from Assumption 2 we have that $D_i$ is i.i.d and $c_i$ is also i.i.d because otherwise, the user will be able to gauge other user's payment.
>
> >**W4: The "symmetric threshold strategy" is used in the main text without definition (and without even a reference to the appendix!).**
>
> Due to the space limit, we postpone it to  Definition 4 (threshold strategy) of Section.A.1 in the Appendix. In game theory, a symmetric threshold strategy might involve players adopting the same threshold for their strategies. We will change the "threshold strategy" to the "symmetric threshold strategy" and move it to the main paper if space allows.
>
> >**W5: The paper makes a common ground-truth assumption (same
> ). In practice, this may be unrealistic.**
>
> We admit that a common underlying parameter $\theta^*$ may not be the case in reality. However, in the statistical estimation literature, this is quite normal as we always assume the data are i.i.d. sampled from the same distribution or model. Moreover, current research on truthful and private mechanism design [1, 2] and private statistical estimation [3-5] always need such an i.i.d. assumption. Thus, such an assumption is reasonable.
>
> [1] Cummings, Rachel, Stratis Ioannidis, and Katrina Ligett. "Truthful linear regression." Conference on Learning Theory. PMLR, 2015.
>
> [2] Qiu, Yuan, Jinyan Liu, and Di Wang. "Truthful Generalized Linear Models." arXiv preprint arXiv:2209.07815 (2022).
>
> [3] Varshney, Prateek, Abhradeep Thakurta, and Prateek Jain. "(Nearly) Optimal Private Linear Regression via Adaptive Clipping." arXiv preprint arXiv:2207.04686 (2022).
>
> [4] Cai, T. Tony, Yichen Wang, and Linjun Zhang. "The cost of privacy: Optimal rates of convergence for parameter estimation with differential privacy." The Annals of Statistics 49.5 (2021): 2825-2850.
>
> [5] Bassily, Raef, et al. "Private stochastic convex optimization with optimal rates." Advances in neural information processing systems 32 (2019).
>
> >**Q1:  I don't see Assumption 3 as a generalization of Cummings et al. Given that $\delta < 1$
> , the assumption is essentially equivalent to (at least implied by) $f_i < 2c_i \epsilon^3$
> , which is more demanding than $f_i < c_i \epsilon^2$
> . It should be rather presented as a relaxation. Do the authors agree?**
>
> The authors agree with the reviewer's opinion in this regard. Currently, Assumption 3 is indeed not a generalization of the assumption used in the work of Cummings et al. We will delete the generalization part in our paper.
>
> >**Q2: I don't understand the claim (line 162) that "the one-round communication necessitates a closed-form estimator". Wouldn't any algorithm work? If so, I suggest that the authors merely present their new estimator as a proposal for covariance estimation, with a high-probability inversion guarantee (and even point to Theorem 3 as the end goal).**
>
> Here, one-round communication is the protocol that all agents send to the server once and simultaneously. Thus, there are no further interactions, which means not all algorithms work. For example, the based method is not one-round, and it does not work. As there is just one round, the server intuitively needs to aggregate this feedback and get an estimator, which has a closed form.
>
> In the final version, if additional pages are allowed, we will move the results of the high-probability inversion guarantee to the main context.
>
> >**Q3: I fail to understand how $\theta^{*}$ can be recovered without assumptions on the thresholds $r, \tau_x, \tau_y$ and $ \lambda_n$. It seems that this should not be possible for arbitrary values of these parameters. Can the authors clarify this?**
>
> Yes, indeed these parameters cannot be arbitrary and their values need to be carefully decided. Due to the space limit, we introduce how to tune these parameters in Lemma 22 in the Appendix. In the revised version, we will also state these conditions in our Theorem 3 and Corollary 8 in the main paper for ease of understanding.

---

> ### Author Response · Authors · 2024-08-06
> **(Continued) Re: Q4**
>
> >**Q4: Right now, though I think I mostly understand the algorithm, and its implications in terms of privacy, strategyproofness and individual rationality, its accuracy is a mystery to me, which prevents me from providing a larger score. I would be greatly thankful to the authors if they could help me gain insights into this aspect of their results.**
>
> We thank the reviewer for appreciating our contributions. We present the accuracy result in Theorem 3. In the upper bound of Thm.3 there are two terms. The first one is $\tilde{O}(\frac{\alpha^2  n k}{\epsilon^2})$ (we assume the radius $r$ as a constant), where $\alpha$ is defined in Definition 6 is $1-\alpha$. Note that if all agents' cost is less than the threshold, then this term becomes zero. Moreover, as $1-\alpha$ means our participation goal, by choosing the parameter $\alpha$ sufficiently small (as suggested in Corollary 8), it will be dominated by the second term.
>
> The second term is $\tilde{O}(\frac{k}{n\epsilon^2})$, which is the cost due to privacy and statistical error. It is notable that even in the standard (non-private) high dimensional sparse linear model, the optimal estimation error is $\tilde{O}(\frac{k}{n})$, which only depends on $\log d$ and the sparsity. As we consider the high dimensional case where $n\ll d$, this bound will be $o(1)$. Thus, our second term is also very small and comparable to the non-private case.

---

> > ### Comment · Reviewer_81kj · 2024-08-09
> >
> > I thank the authors for their response, which clarified several points. Overall, I believe that the paper would greatly gain from improved writing, which is why I stick with my rating. But I fully recognize the value of their contribution.

---

> > > ### Author Response · Authors · 2024-08-10
> > >
> > > We appreciate the reviewer's recognition and feedback. For the camera-ready version, we will incorporate the requested clarifications and definitions on the additional page to ensure completeness. We will also correct any typos and unify the notation to enhance readability. If our response has addressed your concerns and positively influenced your view of our paper, we kindly ask you to consider revising the score.

---

### Official Review · Reviewer_BxFH · 2024-07-07

**Soundness:** 3
**Presentation:** 2
**Contribution:** 2
**Rating:** 3
**Confidence:** 4

**Summary:**

The paper solves the problem of high dimensional sparse regression with subgaussian covariates. Along with doing that they also ensure differential privacy of the data providers. Finally, they also provide a payment scheme that is individually rational and which incentivizes truthfulness.

**Strengths:**

The paper provides an algorithm to solve the problem of high dimensional sparse regression with subgaussian covariates. They also make sure that the algorithm also ensures differential privacy of the data providers. Finally, they also provide a payment scheme that is individually rational and which incentivizes truthfulness.

**Weaknesses:**

1) The paper seems to be a slight extension of differentially private logistic regression while also incorporating payments into it which does not seem like a major extension given the earlier works of Fallah et al. or Anjarlekar et al. have incorporated incentive-compatible payment mechanisms in traditional DP settings with Fallah et al. solving the problem for mean estimation while Anjarlekar et al. extending that work for the case of logistic regression.
2) The assumptions made in the paper do not seem practical (for example assumptions 3 and 4)

**Questions:**

1) It is unclear to me why the authors have used Joint Differential Privacy instead of the original DP definition. Can the authors provide more justification for this?
2) Assumption 3 seems a bit vague. Can the authors clarify the specific structure of the upper bound used in the assumption? It seems more clear to have f(.) which is a monotonically increasing convex function of $\epsilon$.
3) Can the authors clarify more about the practical validity of Assumption 4?
4) A literature survey related to incorporating payments in differentially private models seems incomplete. Some relevant works are as follows
[a] Justin Kang, Ramtin Pedarsani, & Kannan Ramchandran. (2024). The Fair Value of Data Under Heterogeneous Privacy Constraints in Federated Learning.
[b] Ameya Anjarlekar, Rasoul Etesami, & R. Srikant. (2023). Striking a Balance: An Optimal Mechanism Design for Heterogenous Differentially Private Data Acquisition for Logistic Regression.
5) Can be approach be extended to scenarios incorporating heterogeneous differential privacy? Similarly can the proposed approach work for larger models such as deep neural networks?
6) It would have been better to add some experimental results to highlight how the payments and model error varies with a change in the differential privacy guarantees and other parameters.

---

> ### Author Rebuttal · Authors · 2024-08-06
>
> >**W1: The paper seems to be a slight extension of differentially private logistic regression while also incorporating payments into it which does not seem like a major extension given the earlier works of Fallah et al. or Anjarlekar et al. have incorporated incentive-compatible payment mechanisms in traditional DP settings with Fallah et al. solving the problem for mean estimation while Anjarlekar et al. extending that work for the case of logistic regression.**
>
> ***We respectfully disagree with the reviewer regarding the contribution of our work compared to the work of Fallah et al. [1] or Anjarlekar et al [2]. Importantly, this paper is not a slight extension of DP logistic regression, and it is totally different from the work you mentioned.***
>
> * ***Our proposed DP algorithm is not adapted from DP logistic regression or mean estimation literature [1,2].*** The work of Anjarlekar et al. [2] is based on the previous literature on objective perturbation, which is an iterative algorithm. And the work of Fallah et al. [1] is based on adding Laplacian to a weighted mean. Compared to them, we developed a new closed-form estimator for sparse linear regression.
>
> * ***The methods you mentioned cannot solve our problem.*** Specifically, it is unknown where the objective perturbation method is suitable for high-dimension sparse linear regression, even if we only consider DP, as the utility will always depend on the dimensionality. the work of Fallah et al. [1] is based on adding Laplacian to a weighted mean, which obviously cannot solve our problem.
>
> * We provided theoretical results on individual rationality, Bayesian Nash equilibrium, and the payment budget, which have not been given by previous work Anjarlekar et al. [2].
>
> * While there has been work on incorporating IC payment scheme into DP setting, we want to highlight that ***our paper are the first work to study in the high dimensional setting.*** High dimensionality gives rise to several consequences: (1) the regularization techniques used by prior work are not applicable, (2) a (novel) covariance matrix estimator is needed to guarantee the invertibility, (3) this assumption precludes the use of the output perturbation mechanism adopted in the previous work and therefore we seek to privatize by sufficient perturbation scheme, which in turn greatly makes our truthful analysis part more complicated. We strongly recommend the reviewer to go through Section.C in the Appendix for a comprehensive understanding and accurate evaluation of our work.
>
> [1] Fallah, Alireza, Ali Makhdoumi, Azarakhsh Malekian, and Asuman Ozdaglar. "Optimal and differentially private data acquisition: Central and local mechanisms." Operations Research 72, no. 3 (2024): 1105-1123.
>
> [2] Anjarlekar, Ameya, Rasoul Etesami, and R. Srikant. "Striking a Balance: An Optimal Mechanism Design for Heterogenous Differentially Private Data Acquisition for Logistic Regression." arXiv preprint arXiv:2309.10340 (2023).
>
> >**W2: The assumptions made in the paper do not seem practical (for example assumptions 3 and 4)**
>
> We made five assumptions throughout the paper.
> - Assumption 1 is the boundedness of the underlying parameter $\theta^*$ and the covariance matrix.  As we mentioned in the paper, such assumptions have also been used in existing literature.
> - Assumption 2 is sub-Gaussianity on the covariate vector and response. Note that such an assumption is natural in the literation of statistical estimation and is weaker than the boundness assumption in [1, 2].
> - Assumption 3 is a stronger version of [1]. We mimic their assumption on upper bounding the privacy cost function but instead of quadratic bound, we use cubic bound. This is a bit stronger since our assumption on the distribution of response is more relaxed than prior work and we consider the high dimensional sparse setting. Note that the quadratic bound assumption for $(\epsilon,\delta)$-DP is reasonable, as mentioned in Appendix D of [1]. ***We would like to point out that the upper bound on the privacy cost function is necessary for truthfulness analysis and it is also reasonable to assume that strategic users have bounded privacy cost functions.*** We will leave the problem of how to relax the assumption in future work.
> - Assumption 4 is the conditional independence of privacy cost coefficient $c_i$ of a user $i$ and the data of the other user's data $D_{-i}$ and cost coefficient $c_{-i}$ given a user's data $D_i$. ***This is a practical assumption since in our setting users are also concerned that their payment might be inferred from the privacy cost coefficient.*** This also means $c_i$ does not reveal any additional information about the costs or data of any other users.
> - Assumption 5 is the exponential tail decay of $c_i$ which we adopt from [1, 2].
>
> We believe that each of these assumptions is necessary to carry out the analysis and also reasonable in practice. Please see below for more explanation about Assumption 3 and 4.
>
> [1] Cummings, Rachel, Stratis Ioannidis, and Katrina Ligett. "Truthful linear regression." Conference on Learning Theory. PMLR, 2015.
>
> [2] Qiu, Yuan, Jinyan Liu, and Di Wang. "Truthful Generalized Linear Models." arXiv preprint arXiv:2209.07815 (2022).

---

> ### Author Response · Authors · 2024-08-06
> **(Cont'd) Response to Reviewer BxFH' s questions**
>
> >**Q1: It is unclear to me why the authors have used Joint Differential Privacy instead of the original DP definition. Can the authors provide more justification for this?**
>
> Full differential privacy requires that all outputs by the mechanism, including the payment it allocates to a user, are insensitive to every user’s input. In our strategic user setting,  the payment to every user is supposed to be kept secret. Therefore, it is more natural to assume that the payment $\pi_i$ to each user is only observable by the user $i$ while the estimate $\hat{\theta}$ is publicly observable. This consideration exactly falls into the motivation of Joint Differential Privacy.
>
> >**Q2: Assumption 3 seems a bit vague. Can the authors clarify the specific structure of the upper bound used in the assumption? It seems more clear to have $f(.)$ which is a monotonically increasing convex function of $\epsilon$.**
>
> The monotonicity in $\epsilon$ is intuitive: smaller values imply stronger privacy properties. Specifically, $\epsilon = 0 $ indicates the
> output is independent of user $i$’s data. Moreover, we use cubic term in $\epsilon$ to provide stronger constraints. **However, it should be noted that the bounding function $F = c_i (\delta+ 1) \epsilon^3$ is not associated with convexity. In our proof, we also did not use the convexity property.**
>
> >**Q3: Can the authors clarify more about the practical validity of Assumption 4?**
>
> In Assumption 4, we introduce the privacy cost coefficient $c_i$ as a random variable, sampled from some distribution. As mentioned earlier, the payment $\pi_i$ to each user is supposed to be private and the amount of payment is strongly linked to the privacy cost $c_i$. Therefore, it is very natural to assume that $c_i$ depends on each user's own data $D_i = (x_i,y_i)$ and conditioned on $D_i$, $c_i$ does not reveal any additional information on any other user's cost coefficient or data. Otherwise, a strategic user can use his privacy cost coefficient to infer other users' payments.
>
>
> >**Q4: A literature survey related to incorporating payments in differentially private models seems incomplete. Some relevant works are as follows [a] Justin Kang, Ramtin Pedarsani,  Kannan Ramchandran. (2024). The Fair Value of Data Under Heterogeneous Privacy Constraints in Federated Learning. [b] Ameya Anjarlekar, Rasoul Etesami,  R. Srikant. (2023). Striking a Balance: An Optimal Mechanism Design for Heterogenous Differentially Private Data Acquisition for Logistic Regression.**
>
> We thank the reviewer for providing the relevant literature to us. We will add them to the related work section.
>
> >**Q5: Can be approach be extended to scenarios incorporating heterogeneous differential privacy? Similarly can the proposed approach work for larger models such as deep neural networks?**
>
> Sadly, our current framework is not suitable for extending to other scenarios like heterogeneous differential privacy. This is because our privacy guarantee relies on the Billboard Lemma to broadcast the original DP to JDP, and it is unknown whether there is a
> heterogeneous version of Billboard Lemma.
>
> Meanwhile, designing high-dimensional truthful mechanisms with privacy constraints is an
> intrinsically hard problem. Our mechanism cannot be directly applied to larger models such as DNNs like many other truthful mechanisms. Given this, we believe that our work has been a leap and paved the way for future research.
>
> **Please see the above for response to question 6**

---

> > ### Author Response · Authors · 2024-08-10
> >
> > We thank the reviewer for their time and for mentioning the related work which we have not included. We will add these to our paper. If our response has addressed your concerns and positively influenced your view of our paper, we kindly ask you to consider revising the score.

---

> > > ### Comment · Reviewer_BxFH · 2024-08-12
> > >
> > > I am still not convinced about the choice of the bounding function. Do the results still hold if we consider a general monotonic bounding function?

---

> > > > ### Author Response · Authors · 2024-08-13
> > > >
> > > > **Our upper bounding function has to include $c_i, \epsilon$ and $\delta$, where $c_i$ is the privacy cost coefficient.**
> > > >
> > > > | **Paper** | **Upper Bounding Function**          | **Data Assumption** | **Privacy Guarantee**         |
> > > > |-----------|--------------------------------------|---------------------|-------------------------------|
> > > > | Ours      | $c_i(1+\delta) \epsilon^3 $      | Sub-Gaussian        | $(\epsilon, \delta)$-JDP    |
> > > > | [1]       |$  c_i(1+\gamma) \epsilon^4    $    | Sub-Gaussian        | $(\epsilon, \gamma)$-RJDP   |
> > > > | [1]       |$ c_i(1+\gamma) \epsilon^9   $     | Heavy-tailed        | $(\epsilon, \gamma)$-RJDP   |
> > > > | [2]       | $ c_i \epsilon^2     $             | Bounded       |$ \epsilon$-JDP            |
> > > >
> > > > It is clear that due to different data settings, our assumption is weaker than that of [1] but is stronger than the assumption in [2] which was first introduced by [3] and later adopted by [4].
> > > >
> > > > To see why the upper bounding function in [2] is quadratic in $\epsilon$, we additionally define $g_i$ as the privacy cost for reporting $(x_i
> > > > , y_i)$ to the mechanism $M$ when all other agents report $(X_{-i}, y_{−i})$ and the output is $\hat{\theta}$.
> > > >
> > > > In [3], the authors assume that
> > > >
> > > > $g_i\left(M, \hat{\theta},\left(x_i, y_i\right),\left(X_{-i}, y_{-i}\right)\right) \leq c_i \ln \left(\max_{y_i^{\prime}, y_i^{\prime \prime}} \frac{\operatorname{Pr}\left[M\left(X, y_i^{\prime}, y_{-i}\right)=\hat{\theta}\right]}{\operatorname{Pr}\left[M\left(X, y_i^{\prime \prime}, y_{-i}\right)=\hat{\theta}\right]}\right)$.
> > > >
> > > > This assumption is natural because if $g_i$ is small if the agent $i$'s report exerts little influence on $\hat{\theta}$ and therefore should be upper bounded by a function that depends on the effect that the agent $i$’s report has on the mechanism’s output.
> > > >
> > > > And by Composition Lemma (Lemma 3.2 in [3]),
> > > >
> > > > $\mathbb{E}\left[g_i\left(M, M(X, y),\left(x_i, y_i\right),\left(X_{-i}, y_{-i}\right)\right)\right]-\mathbb{E}\left[g_i\left(M, M\left(X, y_i^{\prime}, y_{-i}\right),\left(x_i, y_i\right),\left(X_{-i}, y_{-i}\right)\right)\right] \leq 2 c_i \epsilon\left(e^\epsilon-1\right) \leq 4 c_i \epsilon^2$
> > > >
> > > > Therefore by setting the privacy cost function $f_i = \frac{1}{4} \mathbb{E}\left[g_i\left(M, M(X, y),\left(x_i, y_i\right),\left(X_{-i}, y_{-i}\right)\right)\right]$, it can be interpreted as the agent $i$’s expected cost for participating in the mechanism up to a scaling constant.
> > > >
> > > > Given that our assumption regarding $f_i$ stems from natural ground and has been used widely in similar settings, we argue that it is a reasonable assumption.
> > > >
> > > > [1] Qiu, Yuan, Jinyan Liu, and Di Wang. "Truthful Generalized Linear Models." arXiv preprint arXiv:2209.07815 (2022).
> > > >
> > > >
> > > > [2]  Cummings, Rachel, Stratis Ioannidis, and Katrina Ligett. "Truthful linear regression." Conference on Learning Theory. PMLR, 2015.
> > > >
> > > > [3] Chen, Yiling, et al. "Truthful mechanisms for agents that value privacy." ACM Transactions on Economics and Computation (TEAC) 4.3 (2016): 1-30.
> > > >
> > > > [4] Ghosh, Arpita, et al. "Buying private data without verification." Proceedings of the fifteenth ACM conference on Economics and computation. 2014.

---

> > > > ### Author Response · Authors · 2024-08-14
> > > >
> > > > Dear Reviewer BxFH,
> > > >
> > > > Please see above our detailed response to your question regarding our assumption 3. We believe that we have used a very general and practical bounding function. If you think our response clarifies your concerns, we kindly ask you to consider raising the score.
> > > >
> > > > Best Regards,
> > > > Authors

---

### Official Review · Reviewer_YR32 · 2024-07-13

**Soundness:** 3
**Presentation:** 3
**Contribution:** 3
**Rating:** 7
**Confidence:** 3

**Summary:**

This paper focuses on mechanism design that incentivizes truthful data reporting while preserving privacy in the context of high dimensional sparse linear regression. The proposed mechanism is $(o(1),O(n^{-Omega(1)}))$-jointly differentially private, provides an estimator that is $o(1)$ accurate, is an approximate Bayes NE where most of the agents report truthfully, asymptotically individually rational, and requires a small payment budget.

**Strengths:**

1. This paper has a very clean presentation despite the complicated problem setting and technical components.
2. The theoretical guarantees are comprehensive, including estimation error, privacy guarantee, truthfulness, individual rationality, and budget.
3. The private estimator is quite interesting.

**Weaknesses:**

Currently, each agent only collects one data point. Is it possible to study the free-riding issue as well under the current framework?

**Questions:**

See above

**Limitations:**

See above

---

> ### Author Rebuttal · Authors · 2024-08-06
>
> We thank the reviewer for posing an interesting question. In our setting, we need to assume each user can only manipulate the response. Thus, it is unclear at this point whether our payment scheme can tackle the free-rider issue. We will leave it as a future work.

---

> > ### Comment · Reviewer_YR32 · 2024-08-12
> >
> > Thank you for your reply. I have no more questions.

---

### Author Rebuttal · Authors · 2024-08-06

**Response to Reviewer BFxH's question 6 about adding experiment**

>**Q6: It would have been better to add some experimental results to highlight how the payments and model error varies with a change in the differential privacy guarantees and other parameters.**

We wish to underscore that the essence of our contribution is theoretical. We would also like to bring the reviewer's attention to the broader field of truthful mechanism design where the majority does not have experiments. Some preliminary experiment results are provided now and more results will come out in our finalized version.

This figure plots the results of our mechanism. It is clear from the figure that when the sample size increases, the error goes down for every value of $\epsilon$. Different privacy budget values $\epsilon$ do make a great difference in error under a small data size regime. When the data size becomes larger, all errors quickly reduce to a very low level. This matches our estimation error $\tilde{O}(\frac{\sqrt{k}}{\sqrt{n}\epsilon})$.

---

### Decision · Program_Chairs · 2024-09-25

**Decision:**

Accept (poster)

**Comment:**

Two paper reviews are positive, and one negative at a reject.

However, as an expert in both differential privacy and mechanism design, and after taking a look at the paper myself, I should note that I strongly disagree with the negative assessment of Reviewer BxFH. Reviewer BxFH's review seems to conflate and confuse several differential privacy and mechanism design and complain about lack of novelty in comparison to work that is only related at a superficial level, in that it includes the keywords "differential privacy", "regression", and "mechanism design". Reviewer BxFH also largely ignored author responses, as well as my own comment, and never replied to the last comment trying to clarify the issues with the paper. I think Review BxFH provides largely insufficient evidence to reject this paper.

I recommend to accept the paper as a poster.